# Structural basis of Gabija anti-phage defence and viral immune evasion

Sadie P. Antine[1,2], Alex G. Johnson[1,2], Sarah E. Mooney[1,2], Azita Leavitt[3], Megan L. Mayer[4], Erez Yirmiya[3], Gil Amitai[3], Rotem Sorek[3] & Philip J. Kranzusch[1,2,5 ✉]

Bacteria encode hundreds of diverse defence systems that protect them from viral infection and inhibit phage propagation[1–5]. Gabija is one of the most prevalent anti-phage defence systems, occurring in more than 15% of all sequenced bacterial and archaeal genomes[1,6,7], but the molecular basis of how Gabija defends cells from viral infection remains poorly understood. Here we use X-ray crystallography and cryo-electron microscopy (cryo-EM) to define how Gabija proteins assemble into a supramolecular complex of around 500 kDa that degrades phage DNA. Gabija protein A (GajA) is a DNA endonuclease that tetramerizes to form the core of the anti-phage defence complex. Two sets of Gabija protein B (GajB) dimers dock at opposite sides of the complex and create a 4:4 GajA–GajB assembly (hereafter, GajAB) that is essential for phage resistance in vivo. We show that a phage-encoded protein, Gabija anti-defence 1 (Gad1), directly binds to the Gabija GajAB complex and inactivates defence. A cryo-EM structure of the virally inhibited state shows that Gad1 forms an octameric web that encases the GajAB complex and inhibits DNA recognition and cleavage. Our results reveal the structural basis of assembly of the Gabija anti-phage defence complex and define a unique mechanism of viral immune evasion.

Bacterial Gabija defence operons encode the proteins GajA and GajB, which together protect cells against diverse phages[1]. To define the structural basis of Gabija anti-phage defence, we co-expressed *Bacillus cereus* VD045 GajA and GajB and determined a 3.0 Å X-ray crystal structure of the protein complex (Fig. 1a,b, Extended Data Fig. 1a,b and Extended Data Table 1). The structure of the GajAB complex reveals an intricate 4:4 assembly with a tetrameric core of GajA subunits braced on either end by dimers of GajB (Fig. 1b). We focused our analysis first on individual Gabija protein subunits. GajA contains an N-terminal ATPase domain that is divided into two halves by the insertion of a protein dimerization interface (discussed further below) (Fig. 1c). The GajA ATPase domain consists of an eleven-stranded β-sheet (β1[ABC], β2[ABC], β4–6[ABC] and β3[ABC], β7–11[ABC]) that folds around the central α1[ABC] helix (Fig. 1c and Extended Data Fig. 2a). Sequence analysis of diverse GajA homologues shows that the GajA ATPase domain contains a highly conserved ATP-binding site that is shared with canonical ABC ATPase proteins[8] (Extended Data Fig. 2a). The GajA C terminus contains a four-stranded parallel β-sheet β1–β4[T] surrounded by three α-helices α3[T], α4[T] and α12[T] that form a Toprim (topoisomerase-primase) domain associated with proteins that catalyse double-stranded DNA (dsDNA) breaks[9,10] (Fig. 1c and Extended Data Fig. 2a). Consistent with a role in dsDNA cleavage, the structure of GajA confirms previous predictions of overall shared homology between GajA and a class of DNA endonucleases named OLD (overcoming lysogenization defect) nucleases[11,12]. Discovered at first as an *Escherichia coli* phage P2 protein responsible for cell toxicity in

*recB* and *recC* mutant cells[13–15], OLD nucleases occur in diverse bacterial genomes, either as single proteins (class 1) or associated with partner UvrD/PcrA/Rep-like helicase proteins (class 2), but the specific function of most OLD nuclease proteins is unknown[11,12]. GajA is a class 2 OLD nuclease, with the Toprim domain containing a complete active site composed of DxD after β3[T] (D432 and D434), an invariant glutamate after β2[T] (E379) and an invariant glycine between α1[T] and β1[T] (G409). This is similar to the active site of *Burkholderia pseudomallei* (*Bp*OLD), which was previously shown to be essential for a two-metal-dependent mechanism of DNA cleavage[11] (Fig. 1d and Extended Data Fig. 2a).

The structure of GajB reveals a superfamily 1A DNA helicase domain. Bacterial DNA helicases belonging to this superfamily typically have a role in DNA repair[16] (Fig. 1e). Superfamily 1A helicase proteins such as UvrD, Rep and PcrA translocate along single-stranded DNA in the 3′ to 5′ direction, and are architecturally divided into four subdomains—1A, 1B, 2A and 2B—that reposition relative to each other during helicase function[16]. GajB contains all of the conserved helicase motifs that are required for ATP hydrolysis and nucleic acid unwinding, including a Walker A motif Gx(4)GK-[TT] and a UvrD-like DEXQD-box Walker B motif that is responsible for the hydrolysis of nucleoside triphosphate[16–18] (Fig. 1f and Extended Data Fig. 3a). Activation of superfamily 1A DNA helicase proteins such as UvrD and Rep is known to require protein dimerization and the rotation of the 2B subdomain[19–21]. Comparisons with UvrD and Rep show that GajB protomers in the GajAB complex exhibit a partial rotation of the 2B domain relative to 2A–1A–1B,

[1]Department of Microbiology, Harvard Medical School, Boston, MA, USA. [2]Department of Cancer Immunology and Virology, Dana-Farber Cancer Institute, Boston, MA, USA. [3]Department of Molecular Genetics, Weizmann Institute of Science, Rehovot, Israel. [4]Harvard Center for Cryo-Electron Microscopy, Harvard Medical School, Boston, MA, USA. [5]Parker Institute for Cancer Immunotherapy at Dana-Farber Cancer Institute, Boston, MA, USA. ✉e-mail: philip_kranzusch@dfci.harvard.edu

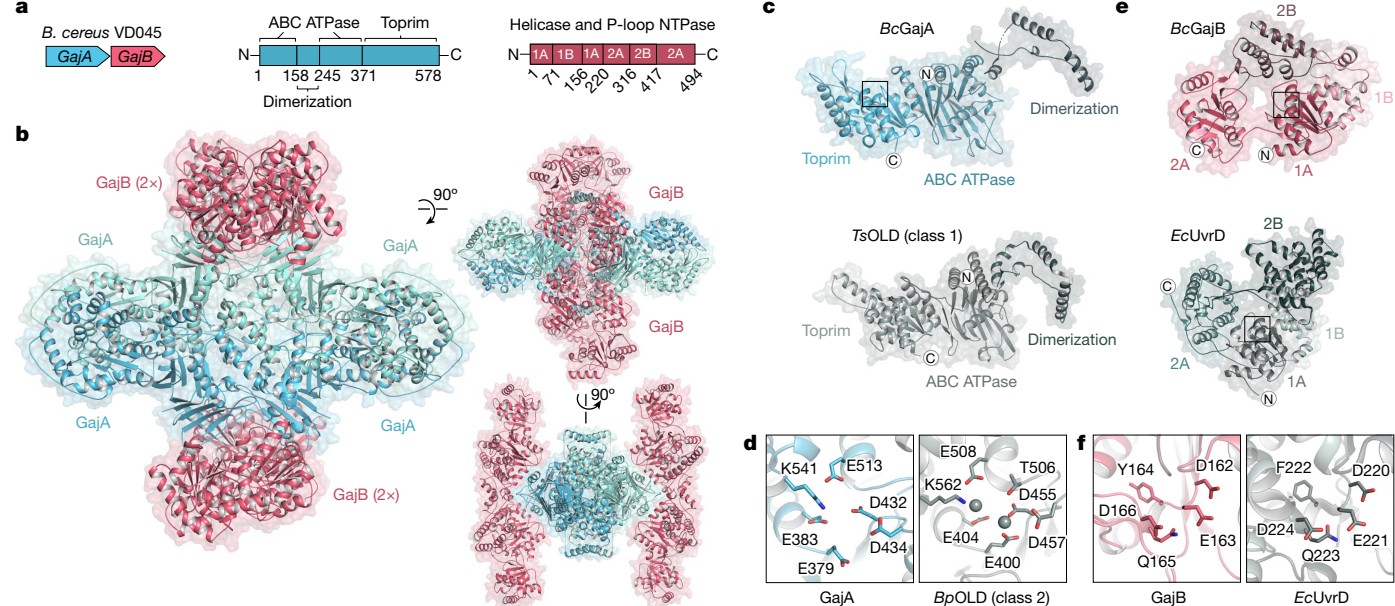

**Fig. 1 | Structure of the Gabija anti-phage defence complex. a**, Schematic of *B. cereus* (*Bc*) Gabija defence operon and domain organization of GajA and GajB. **b**, Overview of the GajAB X-ray crystal structure shown in three orientations. GajA protomers are depicted in two shades of blue and GajB protomers are in red. **c**, Isolated GajA monomer (top) and comparison with a *Ts*OLD nuclease monomer (bottom) (Protein Data Bank (PDB) ID:6P74)[12]. **d**, Magnified views of Toprim catalytic residues in GajA (left) and *Bp*OLD (right) (PDB ID: 6NK8)[11]. The location of the GajA cut-away image is indicated with a box in **c** and magnesium ions are depicted as grey spheres. **e**, Isolated GajB monomer (top) and comparison with *E. coli* (*Ec*) UvrD (bottom) (PDB ID: 2IS2)[20]. **f**, Magnified views of the DEXQD-box motif in GajB (left) and *Ec*UvrD (right). The locations of the GajB and UvrD cut-away images are indicated with boxes in **e**.

consistent with a partially active conformation that is poised to interact with phage DNA (Extended Data Fig. 1e).

## Gabija forms a supramolecular complex

To define the mechanism by which the Gabija complex assembles, we analysed oligomerization interfaces within the GajAB structure. Purification of individual Gabija proteins shows that GajA alone is sufficient to oligomerize into a homo-tetrameric assembly (Extended Data Fig. 1a). GajB migrates as a monomer on size-exclusion chromatography, supporting a stepwise model of GajAB assembly (Fig. 2a and Extended Data Fig. 1a). GajA tetramers form through two highly conserved oligomerization interfaces (Fig. 2b,c and Extended Data Fig. 2). First, the GajA N-terminal ATPase domain contains an insertion between β7[ABC] and β8[ABC] that consists of four α-helices (α1–α4[D]) that zip up against a partnering GajA protomer to form a hydrophobic interface along the α2[D] helix (Fig. 2b). A similar α1–α4[D] dimerization interface exists in the structure of the *Thermus scotoductus* class 1 OLD (*Ts*OLD) protein, which shows that this interface is conserved within divergent OLD nucleases[12] (Figs. 1c and 2c). The GajA ATPase domain contains a second oligomerization interface in a loop between β6[ABC] and α6[ABC], in which hydrogen-bond contacts between D135 and R139 interlock two GajA dimers to form the tetrameric core assembly (Fig. 2c). Compared to GajA, the GajB–GajB dimerization interface is minimal and consists of a hydrophobic surface in the GajB helicase 1B domain centred at Y119 and I122 (Fig. 2d). Major GajA–GajB contacts also occur in the GajB helicase 1B domain, in which GajA R97 in a loop between α4[ABC] and β5[ABC] forms hydrogen-bond contacts with Q150 in GajB α7 along with hydrophobic packing interactions centred at GajB V147 (Fig. 2e and Extended Data Fig. 3a). Notably, the GajAB structure shows that the GajB helicase 1A subdomain, which includes the DEXQD-box active site, is positioned adjacent to the GajA ATPase domain, suggesting that GajB ATP hydrolysis and DNA-unwinding activity might regulate the activation of the GajA ATPase domain (Fig. 2e).

In addition to the major GajAB interface contacts, Gabija supramolecular complex assembly is driven by extensive protomer interactions that result in around 31,000 Å² of surface area buried for the GajA tetramer and around 1,800 Å² of surface area buried for each GajB subunit.

We reconstituted Gabija activity in vitro and observed that the GajAB complex binds to and rapidly cleaves a previously characterized 56-bp dsDNA substrate that contains a sequence-specific motif derived from phage lambda DNA[22] (Extended Data Fig. 1c). The GajAB complex can interact with a scrambled DNA sequence but is unable to cleave this target DNA (Extended Data Fig. 1c,d). GajA and GajB proteins are each essential for phage defence in vivo[1,22], but we observed, in agreement with previous results, that GajA is alone sufficient to cleave target DNA and does not require GajB in vitro[22,23] (Extended Data Fig. 1c). These results suggest that GajAB complex formation could have a specific role in controlling substrate recognition or nuclease activation during phage infection. To confirm these findings, we analysed protein interaction interfaces in the GajAB complex structure and tested the effects of a panel of mutations on the assembly of the Gabija complex in vitro and the ability of Gabija to defend *Bacillus subtilis* cells from phage SPβ infection in vivo. Mutations to the GajA–GajB hetero-oligomerization interface, including GajA K94E and R97A and GajB V147E and Q150R, disrupted complex formation, indicating that these regions are crucial for Gabija complex assembly (Extended Data Fig. 1f). Likewise, these substitutions to the GajA–GajB interface markedly reduced the ability of Gabija to inhibit phage replication in *B. subtilis*. Substitutions to the GajA–GajA dimerization interface including I199E, I212E and K229E also resulted in the complete loss of phage resistance (Fig. 2f). By contrast, phage resistance was tolerant to mutations in the GajB–GajB interface, which suggests that this minimal interaction surface is not strictly essential for anti-phage defence. Together, these results define the structural basis of GajA and GajB interaction and show that the formation of the GajAB supramolecular complex is crucial for Gabija anti-phage defence.

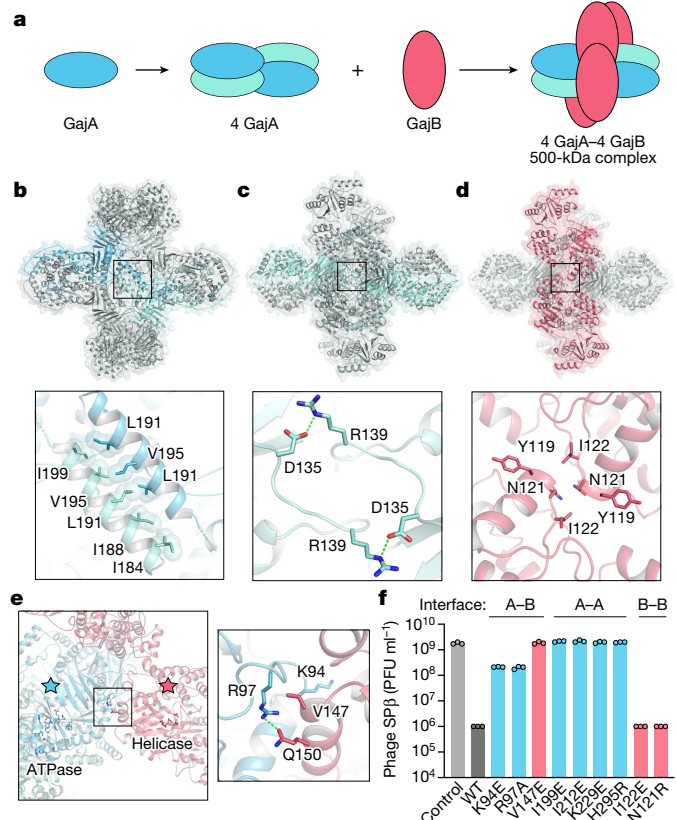

**Fig. 2 | Mechanism of Gabija supramolecular complex assembly. a**, Schematic model of GajAB complex formation by GajA tetramerization and GajB docking. **b**, Overview of the GajA α2$^D$–α2$^D$ dimerization interface and detailed view of interacting residues. For clarity, each GajA monomer is depicted in two shades of blue. **c**, Overview of the GajA–GajA ATPase interaction and detailed view of the inter-subunit D135–R139 interaction. **d**, Overview of the minimal GajB–GajB dimer interface and detailed view of GajB–GajB hydrophobic interactions centred around Y119, N121 and I122. **e**, Left, overview of the GajA–GajB interface, highlighting the proximity of GajA ABC ATPase and GajB helicase active-site residues. Right, the box indicates the location of GajA R97 and GajB V147 and Q150 interaction. **f**, Analysis of mutations in the GajA–GajB (A–B), GajA–GajA (A–A), and GajB–GajB (B–B) multimerization interfaces. GajA and GajB mutations were selected by identifying central residues with well-defined protein–protein contacts within each multimerization interface, and were tested to determine their effects on the ability of the *B. cereus* Gabija operon to defend cells against phage infection. Data represent the phage SPβ average plaque-forming units (PFU) ml$^{-1}$ of three biological replicates, with individual data points shown. WT, wild type.

## Structural basis of Gabija viral evasion

To overcome host immunity, phages encode evasion proteins that specifically inactivate anti-phage defence[24–29]. In a companion study, Yirmiya et al. report the discovery of a viral inhibitor of Gabija anti-phage defence[30], and we reasoned that defining the mechanism of immune evasion would provide further insight into the function of the Gabija complex. Gad1 is a *Bacillus* phage Phi3T protein that is atypically large (35 kDa) compared to other characterized phage immune-evasion proteins (Extended Data Fig. 4a). Protein interaction analysis showed that Gad1 binds directly to GajAB (Extended Data Fig. 4b,c), and we used cryo-EM to determine a 2.6 Å structure of the GajAB–Gad1 co-complex assembly (Fig. 3a,b, Extended Data Figs. 5 and 6a–g and Extended Data Table 2). The GajAB–Gad1 co-complex structure reveals a mechanism of inhibition in which Gad1 proteins form an oligomeric web that wraps 360° around the host defence complex. Eight copies of phage Gad1 encircle the GajAB assembly, forming a 4:4:8 GajAB–Gad1 complex

that is around 775 kDa in size (Fig. 3b,c). Gad1 mainly recognizes the GajA nuclease core, forming extensive contacts along the surface of the GajA dimerization domain (Fig. 3c,d). Key GajAB–Gad1 contacts include hydrogen-bond interactions from a Gad1 positively charged loop located between β5 and β6 with GajA α2$^D$ (Fig. 3e and Extended Data Fig. 7a–c), and hydrophobic packing interactions between Gad1 Y190 and F192 with GajA α2$^D$ (Fig. 3f and Extended Data Fig. 7a). Although the contacts between Gad1 and GajB are limited, both GajA and GajB are necessary for Gad1 interaction, indicating that Gad1 specifically targets the fully assembled GajAB complex to inactivate host anti-phage defence (Extended Data Fig. 4d).

Gad1 wraps around the GajAB complex using a network of homo-oligomeric interactions and notable conformational flexibility. On either side of the GajAB complex, four copies of Gad1 interlock into a tetrameric interface along the primary GajA-binding site (Fig. 3d). The Gad1 tetrameric interface is formed by hydrogen-bond interactions between the C-terminal 'shoulder' domain of each Gad1 monomer and a highly conserved set of three cysteine residues, C282, C284 and C285, which form disulfide interactions at an inter-subunit interface (Fig. 3d,g and Extended Data Fig. 7a,d). The N terminus of each Gad1 monomer forms an 'arm' domain that extends out from the shoulder and reaches around the GajA nuclease active site to connect to a partnering Gad1 protomer from the opposite side of the complex. At the end of the Gad1 arm is an N-terminal 'fist' domain that allows two partnering Gad1 protomers to interact and complete the octameric web assembly (Fig. 3c,h). Structural flexibility limits resolution in this portion of the cryo-EM map, but AlphaFold2 modelling[31,32] and rigid-body placement of the Gad1 N-terminal fist domain suggests that conserved hydrophobic residues around the Gad1 α1 helix mediate the fist–fist interactions (Fig. 3h and Extended Data Fig. 7a). To fully encircle GajAB, Gad1 adopts two distinct structural conformations. Each pair of Gad1 proteins that wrap around and connect at the edge of the GajAB complex are formed by one Gad1 protomer reaching out from the shoulder with an arm domain extended straight down and one Gad1 protomer reaching out with an arm domain bent around 35° to the left (Fig. 3i and Extended Data Fig. 6h). Sequence analysis of Gad1 proteins from phylogenetically diverse phages shows that the Gad1 N-terminal arm domain is highly variable in length (Extended Data Fig. 7a), providing further evidence that conformational flexibility in this region is crucial to inhibit host Gabija defence.

To test the importance of individual GajAB–Gad1 interfaces, we next analysed a series of Gad1 substitution and truncation mutants for their ability to interact with GajAB and inhibit Gabija anti-phage defence. The Gad1 residue F192 is located between β4 and β5 and is part of a highly conserved region that forms the centre of the primary GajA–Gad1 interface (Extended Data Fig. 7a). The Gad1 substitution F192R blocked the ability of Gad1 to interact with GajAB in vitro and inhibit Gabjia anti-phage defence in vivo (Fig. 3j and Extended Data Fig. 7a,e). However, individual mutations throughout the periphery were insufficient to disrupt Gad1 inhibition of Gabjia anti-phage defence. This shows that the large footprint of Gad1 is tolerant to small perturbations that might enable host resistance. Similarly, mutations to the conserved Gad1 cysteine residues in the tetrameric shoulder interface greatly reduced the stability of GajAB–Gad1 complex formation in vitro but only exhibited a threefold decrease and mostly still permitted Gad1 to block phage defence in *B. subtilis* cells (Fig. 3j and Extended Data Fig. 7e). The formation of the GajAB–Gad1 complex was also disrupted in vitro by a Y103R mutation in the Gad1 fist–fist interface (Fig. 3h and Extended Data Fig. 7f). Finally, in contrast to wild-type Gad1, expression of the Gad1 N-terminal fist–arm or C-terminal shoulder domains alone was unable to inhibit Gabija, providing evidence that full wrapping of Gad1 around the GajAB complex is necessary to enable phage evasion of anti-phage defence (Fig. 3j and Extended Data Fig. 7e).

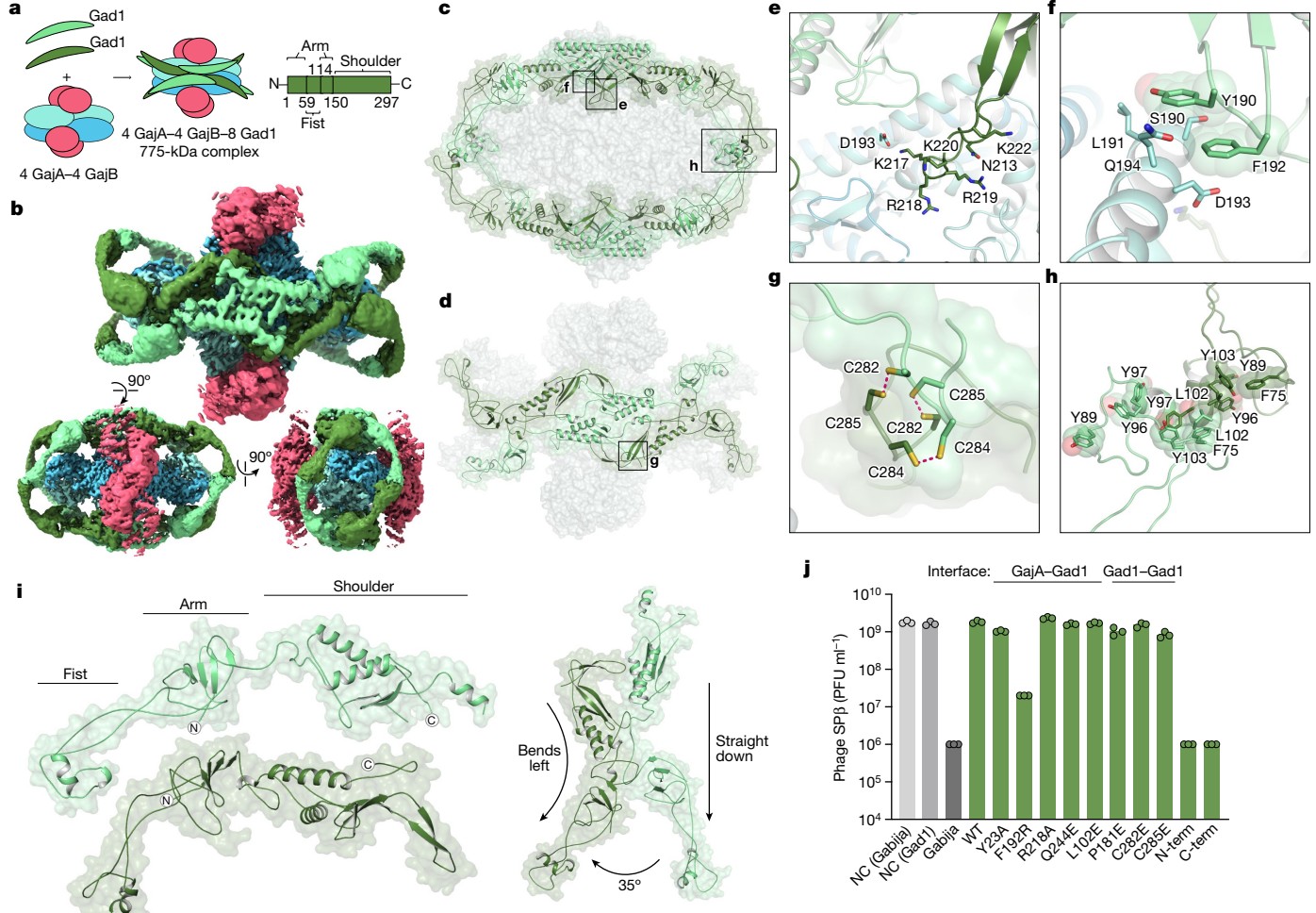

**Fig. 3 | Structural basis of viral evasion of Gabija defence. a**, Schematic model of GajAB–Gad1 co-complex formation and domain organization of phage Phi3T Gad1. **b**, Cryo-EM density map of *Bc*GajAB in complex with Phi3T Gad1, shown in three different orientations. The map is coloured by the model, with Gad1 monomers depicted in two shades of green. **c,d**, Side view of the complete Gad1 octameric complex (**c**) and top-down view of the Gad1 tetrameric interface (**d**), with boxes highlighting views that are magnified in **e**–**h**. **e,f**, Magnified views of major Gad1–GajA interface contacts including a Gad1 positively charged loop (**e**) and hydrophobic interactions with GajA α2$^D$ (**f**). **g,h**, Magnified views of major Gad1–Gad1 oligomerization interactions including disulfide bonds in the C-terminal shoulder domain (**g**) and fist–fist domain contacts modelled by rigid-body placement of an AlphaFold2 fist-domain structure prediction into

the cryo-EM map (**h**). **i**, Two distinct conformations of Gad1 observed in the GajAB–Gad1 co-complex structure. Differences in the rotation of the Gad1 arm domain are highlighted on the right. **j**, Analysis of the effect of Gad1 mutations in the GajA–Gad1 and Gad1–Gad1 multimerization interfaces on the ability of Gad1 to enable evasion of Gabija defence. Data represent PFU ml$^{-1}$ of phage SPβ infecting cells expressing *Bc* Gabija and *Shewanella* sp. phage 1/4 Gad1, or negative control (NC) cells expressing empty vector for either plasmid. *Shewanella* sp. phage 1/4 Gad1 residues are numbered according to the Phi3T Gad1 structure. *Shewanella* sp. phage 1/4 Gad1 N-terminal and C-terminal truncations (N-term and C-term, respectively) are M1–L152 and V153–E348, respectively. Data are the average of three biological replicates, with individual data points shown.

## Gad1 blocks Gabija DNA cleavage

Superposition of the GajAB–Gad1 and GajAB complexes shows that Gad1 binding does not induce a notable conformational change in GajAB, and suggests that Gad1 instead functions through steric hindrance of Gabija anti-phage defence (Extended Data Fig. 7h). To define the mechanism of Gad1 inhibition of Gabija anti-phage defence, we modelled interactions between GajAB and target DNA. The GajA Toprim domain is structurally homologous to the *E. coli* protein MutS, which is involved in DNA repair[33]. Superimposing the MutS–DNA structure revealed positively charged patches lining a groove in the GajA Toprim domain that dips into the nuclease active site (Extended Data Fig. 8). Notably, the Gad1 arm domain directly occupies this putative DNA-binding surface, supporting a model in which the phage protein directly clashes with the path of target dsDNA (Fig. 4a,b). To determine the effect of viral inhibition on GajAB catalytic function, we tested the

role of Gad1 in individual steps of DNA binding and target DNA cleavage. Gad1 prevented GajAB from binding to target DNA and abolished all nuclease activity in vitro (Fig. 4c,d and Supplementary Fig. 1). Gad1 proteins with F192R or C282E mutations were no longer able to inhibit DNA cleavage, in agreement with the inability of F192R-mutant proteins and the reduced ability of C282E-mutant proteins to block Gabija defence in vivo and form stable GajAB–Gad1 complexes in vitro (Extended Data Fig. 7g). Together, these results show that phage Gad1 binds to and wraps around the GajAB complex to block target DNA degradation. Our findings reveal a complete mechanism by which phages evade the Gabija defence system of the host (Fig. 4e).

Our study defines the structural basis of the formation of the Gabija supramolecular complex, and explains how phages block DNA cleavage to overcome this type of host immunity. The approximately 500-kDa GajAB complex expands an emerging theme in anti-phage defence, whereby protein subunits assemble into large machines to resist phage

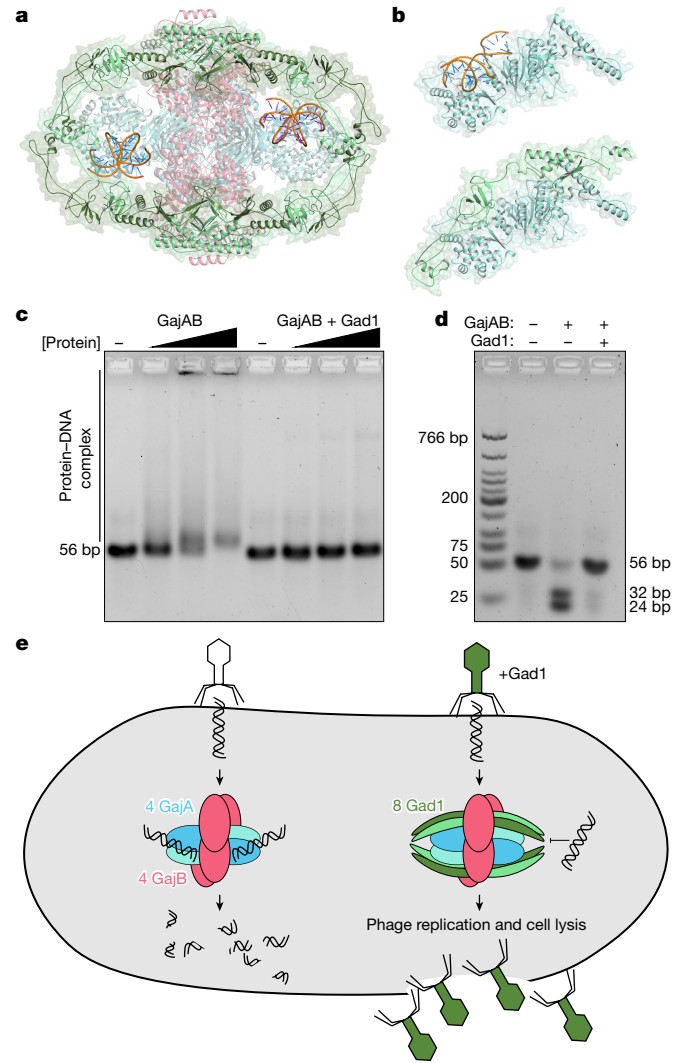

**a**

**b**

**c**
GajAB   GajAB + Gad1
[Protein] –          –

Protein–DNA complex

56 bp

**d**
GajAB: – + +
Gad1: – – +

766 bp

200

75
50

25

56 bp
32 bp
24 bp

**e**

4 GajA

4 GajB

+Gad1

8 Gad1

Phage replication and cell lysis

**Fig. 4 | Inhibition of Gabija DNA binding and cleavage enables viral evasion.**
**a**, Cartoon representation of the GajAB–Gad1 co-complex structure with
modelled DNA (orange), based on structural homology with *E. coli* MutS (PDB
ID 3K0S)[33]. **b**, Top, isolated GajA protomer with modelled DNA (orange) bound
to the Toprim domain. Bottom, the same GajA promoter with Gad1, showing
substantial steric clashes between Gad1 and the path of the DNA. **c**,**d**, Biochemical
analysis of GajAB 56-bp target DNA binding (**c**) and target cleavage (**d**) shows
that Gad1 potently inhibits the activity of GajAB. Data are representative of
three independent experiments. **e**, Model of Gabija anti-phage defence and
mechanism of Gad1 immune evasion.

infection—similarly to the supramolecular complexes in CRISPR[34],
CBASS[35,36] and RADAR[37,38] immunity. These results parallel human innate
immunity, in which key effectors in inflammasome, Toll-like receptor,
RIG-I-like receptor and cGAS–STING signalling pathways also oligomer-
ize into large assemblies to block viral replication[39,40]. In contrast to
the exceptionally large defence complexes of the host, phage evasion
proteins are typically small, 5–20-kDa proteins that sterically occlude
key protein binding and active-site motifs[25,26]. Breaking this rule, the
35-kDa anti-Gabija protein Gad1 is one of the largest described viral
protein–protein inhibitors of host immune signalling (Extended Data
Fig. 4). Whereas most viral evasion proteins that are larger than 20 kDa
are enzymatic domains that catalytically modify target host factors
or signalling molecules, the large size of Gad1 is necessary to bind to,
oligomerize and encircle the entire host GajAB complex. Resistance to
small phage proteins that simply block the GajA active site could explain
why Gabija is a highly prevalent defence system in diverse bacterial

phyla. A key question opened by our structures of the Gabija complex is
how GajB helicase activity is linked to the activation of the GajA nuclease
domain to control the cleavage of target DNA. Gad1 encasing the GajAB
complex to trap it in an inactive state is a new mechanism by which
phages evade host defences, and this finding provides a template to
understand how viruses disrupt the complex mechanisms of activation
of diverse anti-phage defence systems in bacteria.

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

## Methods

### Bacterial strains and phages

*B. subtilis* BEST7003 was grown in MMB (LB supplemented with 0.1 mM $MnCl_2$ and 5 mM $MgCl_2$) with or without 0.5% agar at 37 °C or 30 °C respectively. Whenever applicable, media were supplemented with ampicillin (100 µg ml⁻¹), chloramphenicol (34 µg ml⁻¹) or kanamycin (50 µg ml⁻¹) to ensure the maintenance of plasmids. *B. subtilis* phages phi3T (BGSCID 1L1) and SPβ (BGSCID 1L5) were obtained from the Bacillus Genetic Stock Center (BGSC). Prophages were induced using Mitomycin C (Sigma, M0503).

Phage titre was determined using the small-drop plaque assay method[41]. Four hundred microlitres of overnight culture of bacteria was mixed with 0.5% agar and 30 ml MMB and poured into a 10-cm² plate followed by incubation for 1 h at room temperature. In cases of bacteria expressing Gad1 homologue and Gad1 mutations, 0.1–1 mM IPTG was added to the medium. Tenfold serial dilutions in MMB were performed for each of the tested phages and 10-µl drops were put on the bacterial layer. After the drops had dried up, the plates were inverted and incubated at room temperature overnight. Plaque-forming units (PFUs) were determined by counting the derived plaques after overnight incubation, and lysate titre was determined by calculating PFU ml⁻¹. When no individual plaques could be identified, a faint lysis zone across the drop area was considered to be ten plaques. Efficiency of plating was measured by comparing plaque assay results on control bacteria and bacteria containing the defence system and/or a candidate anti-defence gene.

### Plasmid construction

For protein purification and biochemistry, *B. cereus* VD045 *GajA* (IMG ID 2519684552) and *GajB* (IMG ID 2519684553) genes were codon-optimized for expression in *E. coli*, synthesized as gBlocks (Integrated DNA Technologies) and cloned into custom pET vectors with an N-terminal 6×His-SUMO2 fusion tag (GajB alone) or a C-terminal 6×His tag (GajA alone). GajA and GajB proteins were co-expressed using a custom pET vector with an N-terminal 6×His-SUMO2 or N-terminal 6×His-SUMO2-5×GS tag on GajA and a ribosome-binding site between GajA and GajB. Phi3T and *Shewanella sp.* phage 1/4 Gad1 (IMG ID 2708680195) gBlocks were cloned into a custom pBAD vector containing a chloramphenicol resistance gene and an IPTG-inducible promoter. For Gad1 pull-down assays, *Shewanella sp.* phage 1/4 Gad1 was cloned with a ribosome-binding site after the GajB gene in the N-terminal 6×His-SUMO2-5×GS GajAB plasmid.

For plaque assays, the DNA of Gad1 was amplified from the phage phi3T genome using KAPA HiFi HotStart ReadyMix (Roche, KK2601). Because Gad1 was toxic in *B. subtilis* cells containing Gabija, *Shewanella* sp. phage 1/4 Gad1 was used and synthesized by GenScript. Gad1 and related homologues were cloned into the pSG-thrC-Phspank vector[42] and transformed to DH5α competent cells. The cloned vector and the vector containing Gad1 substitution and truncation mutants were subsequently transformed into *B. subtilis* BEST7003 cells containing Gabija integrated into the *amyE* locus[1], resulting in cultures expressing both Gabija and a Gad1 homologue. As a negative control, a transformant with an identical plasmid containing GFP instead of the anti-defence gene was used. Transformation in *B. subtilis* was performed using MC medium as previously described[1]. Sanger sequencing was then applied to verify the integrity of the inserts and the mutations. The pSG1 plasmids containing point mutations in Gabija were constructed by subcloning the Gabija sequence into pGEM9Z using restriction enzymes, site-directed mutagenesis as previously described[43] and Gibson assembly back into pSG1, and the plasmids were transformed into *B. subtilis* BEST7003 cells. Sanger sequencing of the mutations regions was applied to verify the mutations in Gabija.

### Protein expression and purification

Recombinant GajAB and GajAB–Gad1 complexes were purified from *E. coli* as previously described[44]. In brief, the expression plasmids described above were transformed into BL21(DE3), BL21(DE3)-RIL (Agilent) or LOBSTR-BL21(DE3)-RIL cells (Kerafast), plated on MDG medium plates (1.5% Bacto agar, 0.5% glucose, 25 mM $Na_2HPO_4$, 25 mM $KH_2PO_4$, 50 mM $NH_4Cl$, 5 mM $Na_2SO_4$, 0.25% aspartic acid, 2–50 µM trace metals, 100 µg ml⁻¹ ampicillin and 34 µg ml⁻¹ chloramphenicol) and grown overnight at 37 °C. Five colonies were used to inoculate 30 ml of MDG starter overnight cultures (37 °C, 230 rpm). Ten millilitres of MDG starter cultures were then inoculated in 1 l M9ZB expression cultures (47.8 mM $Na_2HPO_4$, 22 mM $KH_2PO_4$, 18.7 mM $NH_4Cl$, 85.6 mM NaCl, 1% Cas-Amino acids, 0.5% glycerol, 2 mM $MgSO_4$, 2–50 µM trace metals, 100 µg ml⁻¹ ampicillin and 34 µg ml⁻¹ chloramphenicol) and induced with 0.5 mM IPTG after reaching an optical density at 600 nm ($OD_{600\,nm}$) of 1.5 or higher (overnight, 16 °C, 230 rpm).

After overnight induction, cells were pelleted by centrifugation, resuspended and lysed by sonication in 60 ml lysis buffer (20 mM HEPES pH 7.5, 400 mM NaCl, 10% glycerol, 20 mM imidazole and 1 mM DTT). Lysate was clarified by centrifugation, and supernatant was poured over Ni-NTA resin (Qiagen). Resin was then washed with lysis buffer, lysis buffer supplemented with 1 M NaCl and lysis buffer again, and was finally eluted with lysis buffer supplemented with 300 mM imidazole. Samples were then dialysed overnight in 14-kDa MWCO dialysis tubing (Ward's Science) with SUMO2 cleavage by hSENP2 as previously described[29,30]. hSENP2 did not efficiently cleave N-terminal 6×His-SUMO2-GajAB and the complex was therefore purified with an additional 5×GS linker. Proteins for crystallography and cryo-EM were dialysed in dialysis buffer (20 mM HEPES-KOH pH 7.5, 250 mM KCl and 1 mM DTT), purified by size-exclusion chromatography using a 16/600 Superdex 200 column (Cytiva) and stored in gel filtration buffer (20 mM HEPES-KOH pH 7.5, 20 mM KCl and 1 mM TCEP-KOH). Proteins for biochemical assays were dialysed in dialysis buffer, purified by size-exclusion chromatography using a 16/600 Superdex 200 column (Cytiva) or 16/600 Sephacryl 300 column (Cytiva) and stored in gel filtration buffer with 10% glycerol. Purified proteins were concentrated to more than 10 mg ml⁻¹ using a 30-kDa MWCO centrifugal filter (Millipore Sigma), aliquoted, flash-frozen in liquid nitrogen and stored at −80 °C.

Co-expression of Gabija with Phi3T Gad1 results in mild toxicity in *E. coli* grown on MDG medium plates. No toxicity was observed using a closely related Gad1 homologue from the *Shewanella* phage 1/4. Biochemical analysis of Gabija–Gad1 interactions was therefore conducted with *Shewanella* phage 1/4 Gad1. Notably, all Gad1 residues analysed are 100% conserved between Phi3T Gad1 and *Shewanella* phage 1/4 Gad1. For *Shewanella* phage 1/4 Gad1 pull-down assays, SUMO2-5×GS-GajA-GajB-Gad1 point-mutant plasmids were transformed and expressed in BL21(DE3)-RIL or LOBSTR-BL21(DE3)-RIL cells, and subjected to Ni-NTA column chromatography and SUMO2 cleavage with SENP2. Gad1 pull-down was analysed by SDS–PAGE and Coomassie Blue staining.

### Crystallization and X-ray structure determination

Crystals were grown in hanging drop format using EasyXtal 15-well trays (NeXtal). Native GajAB crystals were grown at 18 °C in 2-µl drops mixed 1:1 with purified protein (10 mg ml⁻¹, 20 mM HEPES, 250 mM KCl and 1 mM TCEP-KOH) and reservoir solution (100 mM HEPES-NaOH pH 7.5, 2.4% PEG-400 and 2.2 M ammonium sulfate). Crystals were grown for seven days before cryo-protection with reservoir solution supplemented with 25% glycerol, and were collected by plunging in liquid nitrogen. X-ray diffraction data were collected at the Advanced Photon Source (beamlines 24-ID-C and 24-ID-E). Data were processed using the SSRL autoxds script (A. Gonzalez, Stanford SSRL). Experimental phase information was determined by molecular replacement

using monomeric GajA and GajB AlphaFold2-predicted structures[31,32] in PHENIX[45]. Model building was completed in Coot[22] and then refined in PHENIX. The final structure was refined to stereochemistry statistics as reported in Extended Data Table 1. Structure images and figures were prepared in PyMOL.

### Electrophoretic mobility shift assay
56-bp sequence-specific motif target dsDNA (5′ TTTTTTTTTTTTT TTTTAATAACCCGGTTATTTTTTTTTTTTTTTTTTTTTTTTTTTT 3′) (ref. 22) or scrambled dsDNA (5′ TTTTTTTTTTTTTTTTTTGACAT TACATTCAGTTTTTTTTTTTTTTTTTTTTTTTTTTTTTT 3′) was incubated with a final concentration of 2 μM, 5 μM or 10 μM purified GajAB, GajA[E379A]–GajB or GajAB–Gad1 complexes in 20 μl gel shift reactions containing 1 μM dsDNA, 5 mM $CaCl_2$ and 20 mM Tris-HCl pH 8.0 for 30 min at 4 °C. Ten microlitres was then mixed with 2 μl of 50% glycerol and separated on a 2% TB (Tris-borate) agarose gel. The gel was then run at 250 V for 45 min, post-stained with TB containing 10 μg ml$^{-1}$ ethidium bromide while rocking at room temperature, de-stained in TB buffer for 40 min and imaged on a ChemiDoc MP Imaging System.

### DNA cleavage assay
The same 56-bp dsDNA substrates as above were incubated with GajAB, GajA[E379A]–GajB or GajAB–Gad1 complexes in a 20-μl DNA cleavage reaction buffer containing 1 μM dsDNA, 1 μM GajAB, GajA[E379A]–GajB or GajAB–Gad1, 1 mM $MgCl_2$ and 20 mM Tris-HCl pH 9.0 for 20 min at 37 °C. After incubation, reactions were stopped with DNA loading buffer containing 60 mM EDTA, and 10 μl was analysed on a 2% TB agarose gel, which was run at 250 V for 45 min. The gel was then post-stained while rocking at room temperature with TB buffer containing 10 μg ml$^{-1}$ ethidium bromide, de-stained in TB buffer alone for 40 min and imaged on a ChemiDoc MP Imaging System.

### Cryo-EM sample preparation and data collection
For the SUMO2-GajAB–Gad1 co-complex sample, 3 μl of 1 mg ml$^{-1}$ was vitrified using a Mark IV Vitrobot (Thermo Fisher Scientific). Before sample vitrification, 2/1 Carbon Quantfoil grids were glow-discharged using an easiGlow (Pelco). Grids were then double-sided blotted for 9 s, with a constant force of 0, 100% relative humidity chamber at 4 °C and a 10-s wait time before plunging into liquid ethane and storing in liquid nitrogen.

GajAB–Gad1 co-complex cryo-EM grids were screened using a Talos Arctica microscope (Thermo Fisher Scientific) operating at 200 kV, and the final map was collected on a Titan Krios microscope (Thermo Fisher Scientific) operating at 300 kV. Both microscopes operated with a K3 direct electron detector (Gatan). SerialEM software v.3.8.6 was used for all data collection. For final data collection, a total of 9,243 movies were taken at a pixel size of 0.3115 Å, a total dose of 41.1 e$^-$ per Å$^2$ and a dose per frame of 0.63 e$^-$ per Å$^2$ at a defocus range of −0.8 to −1.9 μm.

### Cryo-EM data processing
SBGrid Consortium provided data-processing software[46]. Movies were imported into cryoSPARC[47] for patch-based motion correction, patch-based CTF estimation, two-dimensional and three-dimensional particle classification and non-uniform refinement. The cryoSPARC data-processing procedure is outlined in Extended Data Fig. 6. In brief, after patch-based CTF estimation, 500 micrographs were selected and autopicked using Blob Picker, which resulted in 625,295 particles after extracting from micrographs. Two-dimensional classifications were then used to generate five templates for Template Picker, from which 110,654 particles were picked from 500 micrographs. After three more rounds of 2D classification, 648,298 particles from all 9,243 micrographs were used in ab initios ($K = 3$), followed by heterogenous refinement. The best class with 573,410 particles was then used to go back and extract from all micrographs, which resulted in 570,485 particles used in a final 2D classification and ab initio. A total of 351,193 particles from

one ab-initio class were used in non-uniform refinement along with defocus and global CTF refinement, resulting in a 2.84 Å $C_1$ symmetry and 2.57 Å $D_2$ symmetry map, which was then used for model building.

### Cryo-EM model building
Model building was performed in Coot[48] by manually docking Alpha-Fold2-predicted structures[31,32] as starting models and then manually completing refinement and model correction. To model the Gad1 fist domain, an AlphaFold2 model of the Gad1 arm–fist region was super-imposed on the cryo-EM density of the manually built shoulder–arm region and then fit into density in Coot[48]. To complete the model for the sparse GajB density, the X-ray GajB structure was superimposed on the cryo-EM density. GajAB–Gad1 model was refined in PHENIX[45], and the structure stereochemistry statistics are reported in Extended Data Table 2. Figures were prepared in PyMOL and UCSF ChimeraX[49].

### Statistics and reproducibility
Experimental details about replicates are found in the figure legends.

### Reporting summary
Further information on research design is available in the Nature Port-folio Reporting Summary linked to this article.

## Data availability
Coordinates and structure factors of the Gabija GajAB complex have been deposited in the PDB under the accession code 8SM3. Coordinates and density maps of the GajAB–Gad1 co-complex are deposited with the PDB and the Electron Microscopy Data Bank (EMDB) under the accession codes 8U7I and EMD-41983. All other data are available in the manuscript or Supplementary Fig. 1. Source data are provided with this paper.

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

**Acknowledgements** We thank J. Asnes, J. Grippen and members of the P.J.K. and R.S. laboratories for comments and discussion, and A. Lu for assistance with X-ray data collection. The work was funded by grants to P.J.K. from the Pew Biomedical Scholars program, the Burroughs Wellcome Fund PATH program, the Mathers Foundation, the Mark Foundation for Cancer Research, the Cancer Research Institute, the Parker Institute for Cancer Immunotherapy and the National Institutes of Health (1DP2GM146250-01), and by grants to R.S. from the European Research Council (ERC-AdG GA 101018520), the Israel Science Foundation (MAPATS grant 2720/22), the Ernest and Bonnie Beutler Research Program of Excellence in Genomic Medicine, the Deutsche Forschungsgemeinschaft (SPP 2330, grant 464312965) and the Knell Family Center for Microbiology. E.Y. is supported by the Clore Scholars Program and in part by the Israeli Council for Higher Education (CHE) via the Weizmann Data Science Research Center. A.G.J. is supported by a Life Science Research Foundation postdoctoral fellowship of the Open Philanthropy Project. X-ray data were collected at the Northeastern Collaborative Access Team beamlines 24-ID-C and 24-ID-E (P30 GM124165), and used a Pilatus detector

(S10RR029205), an Eiger detector (S10OD021527) and the Argonne National Laboratory Advanced Photon Source (DE-AC02-06CH11357). Cryo-EM data were collected at the Harvard Cryo-EM Center for Structural Biology at Harvard Medical School. We thank T. Humphreys for help with cryo-EM data collection. Part of this research was supported by the NIH grant U24GM129547 and was performed at the Pacific Northwest Center for Cryo-EM at Oregon Health & Science University, with access through EMSL (grid.436923.9), a DOE Office of Science User Facility sponsored by the Office of Biological and Environmental Research.

**Author contributions** The study was designed and conceived by S.P.A. and P.J.K. All protein purification and biochemical assays were performed by S.P.A. and S.E.M. Crystallography structural analysis was performed by S.P.A. Cryo-EM structural analysis was performed by S.P.A., A.G.J. and M.L.M. Model building and analysis were performed by S.P.A. and P.J.K. Bioinformatics and protein sequence analysis were performed by E.Y., A.L., G.A. and R.S. Phage challenge assays were performed by A.L. and R.S. Figures were prepared by S.P.A. with assistance from S.E.M. The manuscript was written by S.P.A. and P.J.K. All authors contributed to editing the manuscript, and support its conclusions.

**Competing interests** R.S. is a scientific cofounder and advisor of BiomX and EcoPhage. The remaining authors declare no competing interests.

## Additional information

**Correspondence and requests for materials** should be addressed to Philip J. Kranzusch.

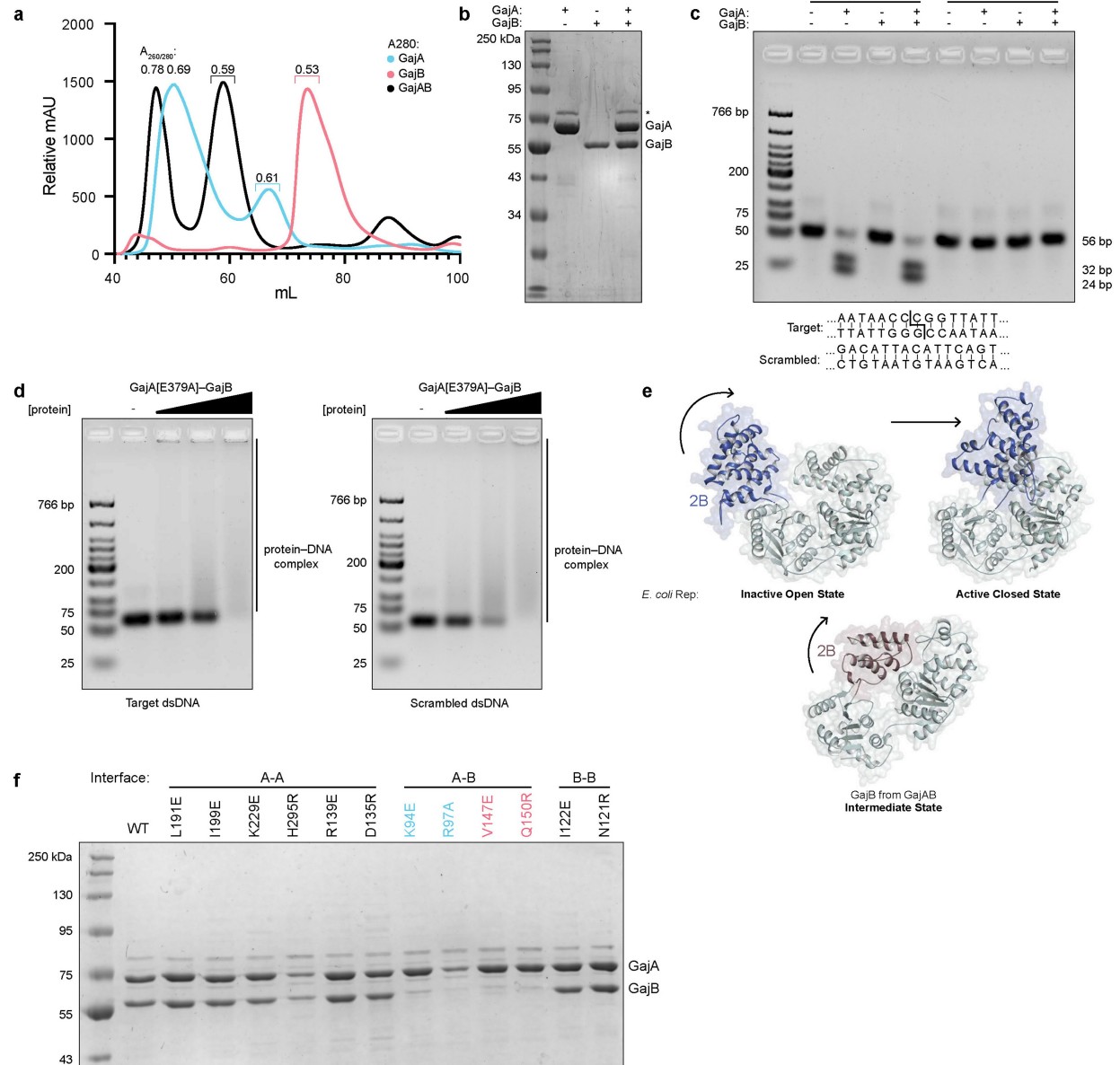

**Extended Data Fig. 1 | GajA and GajB form a supramolecular complex that cleaves phage lambda DNA in vitro. a**, Size-exclusion chromatography (16/600 S200) analysis of recombinant *Bc*GajA and *Bc*GajB proteins, and the co-expressed *Bc*GajAB complex. Brackets indicate fractions collected for biochemical and structural analysis with $A_{260/280}$ of the final purified proteins listed above. **b**, SDS–PAGE analysis of purified GajA, GajB, and GajAB. Asterisk indicates minor contamination with the *E. coli* protein ArnA. Data are representative of at least 3 independent experiments. **c**, Agarose gel analysis of the ability of GajA, GajB, and GajAB to cleave a 56-bp target and scrambled dsDNA demonstrates that GajA alone and the GajAB complex can cleave target DNA only. The sequence-specific GajA target dsDNA with cleavage site

described in Cheng et al.[22] and the scrambled 15-bp sequence are shown below. **d**, Catalytic dead GajA[E379A]–GajB complex binding to target dsDNA (left) and scrambled dsDNA (right). **e**, Structural comparison of GajB and *Ec*Rep (PDB ID 1UAA)[19] demonstrates the GajB 2B domain is rotated in a partially active intermediate position in the GajAB complex structure. **f**, SDS–PAGE analysis of *Bc*GajAB mutant protein complex formation after co-expression and Ni-NTA pull-down demonstrates that mutations to the GajA–GajB interface disrupt complex formation. The GajA and GajB homo-oligomerization interfaces are not required for GajA–GajB interaction, but it is not known if these mutants remain competent at forming the wild-type 4:4 complex. Data in **c**,**d**,**f** are representative of 3 independent experiments.

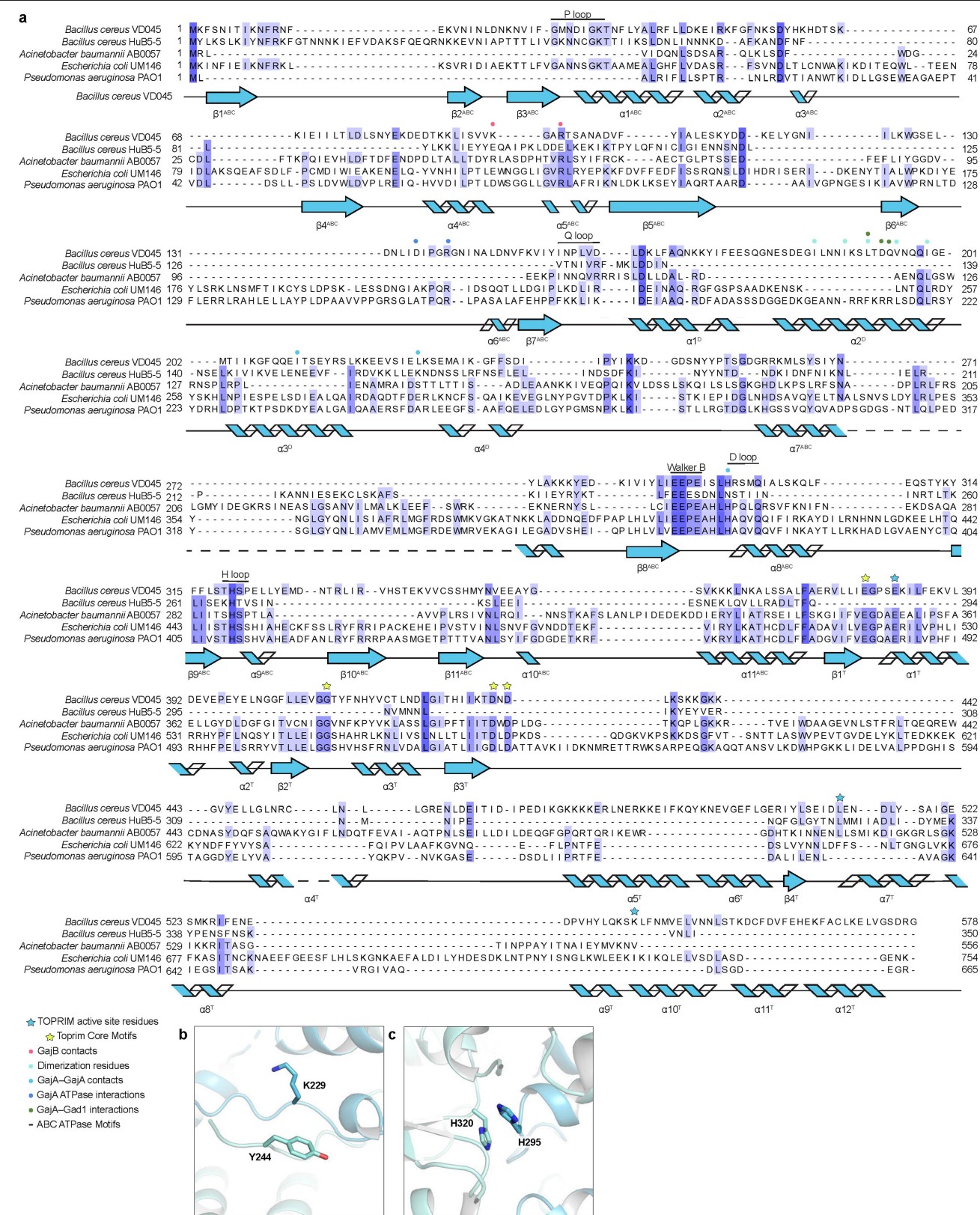

**Extended Data Fig. 2 | Structural characterization of GajA. a**, Structure-guided alignment of GajA proteins from indicated bacteria coloured according to amino acid conservation. The determined *Bacillus cereus* VD045 GajA secondary structure is displayed, and active-site and oligomerization interface residues are annotated according to the key below. Secondary structure abbreviations include ABC ATPase domain (ABC), dimerization domain (D), and Toprim domain (T). **b,c**, Zoomed-in views of GajA–GajA oligomerization interactions including dimerization domain interactions (**b**) and ABC ATPase domain interactions (**c**).

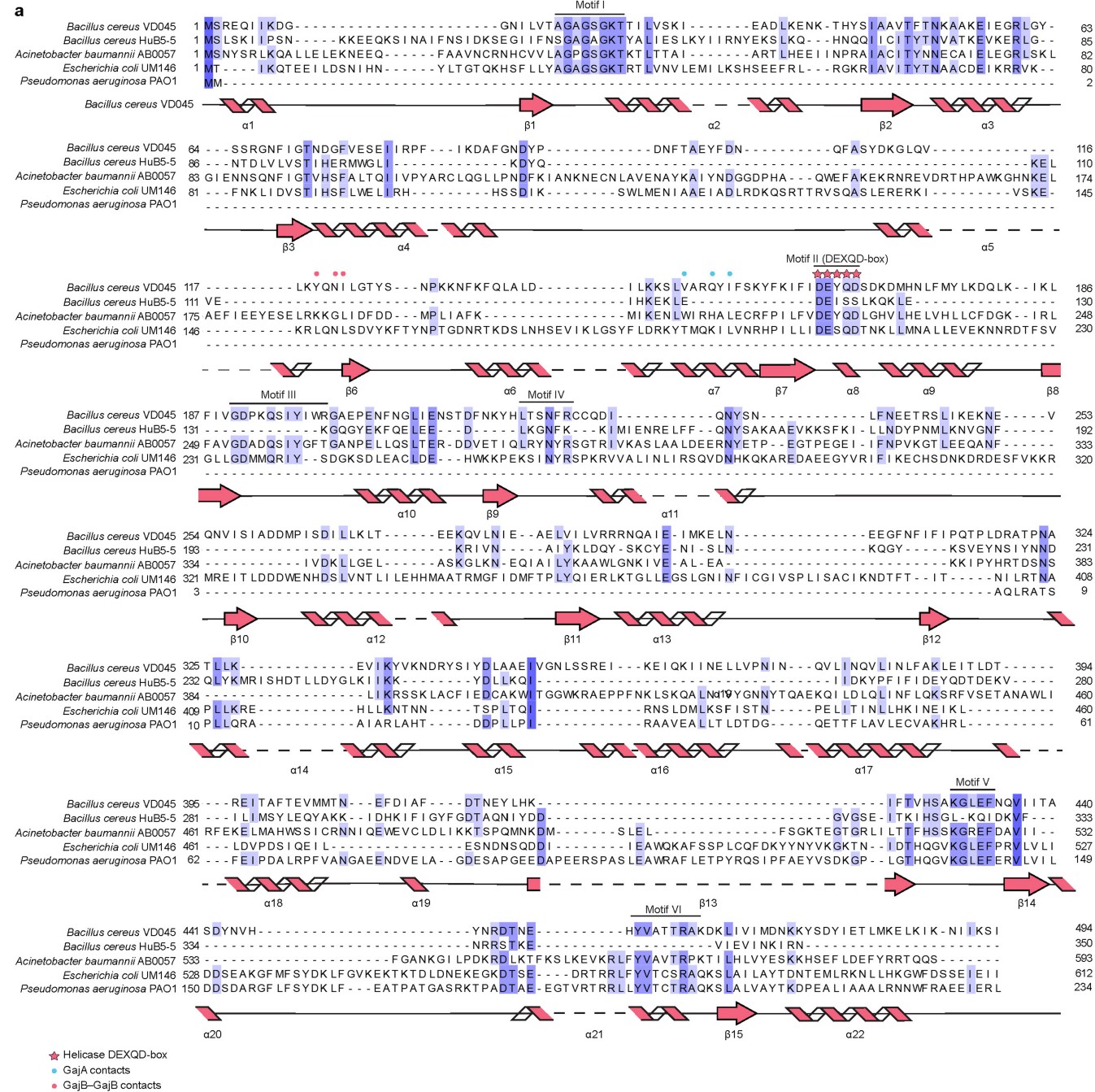

**Extended Data Fig. 3 | Structural characterization of GajB. a**, Structure-guided alignment of GajB proteins from indicated bacteria coloured according to amino acid conservation. The determined *Bacillus cereus* VD045 GajB secondary structure is displayed, and active-site and oligomerization interface residues are annotated according to the key below.

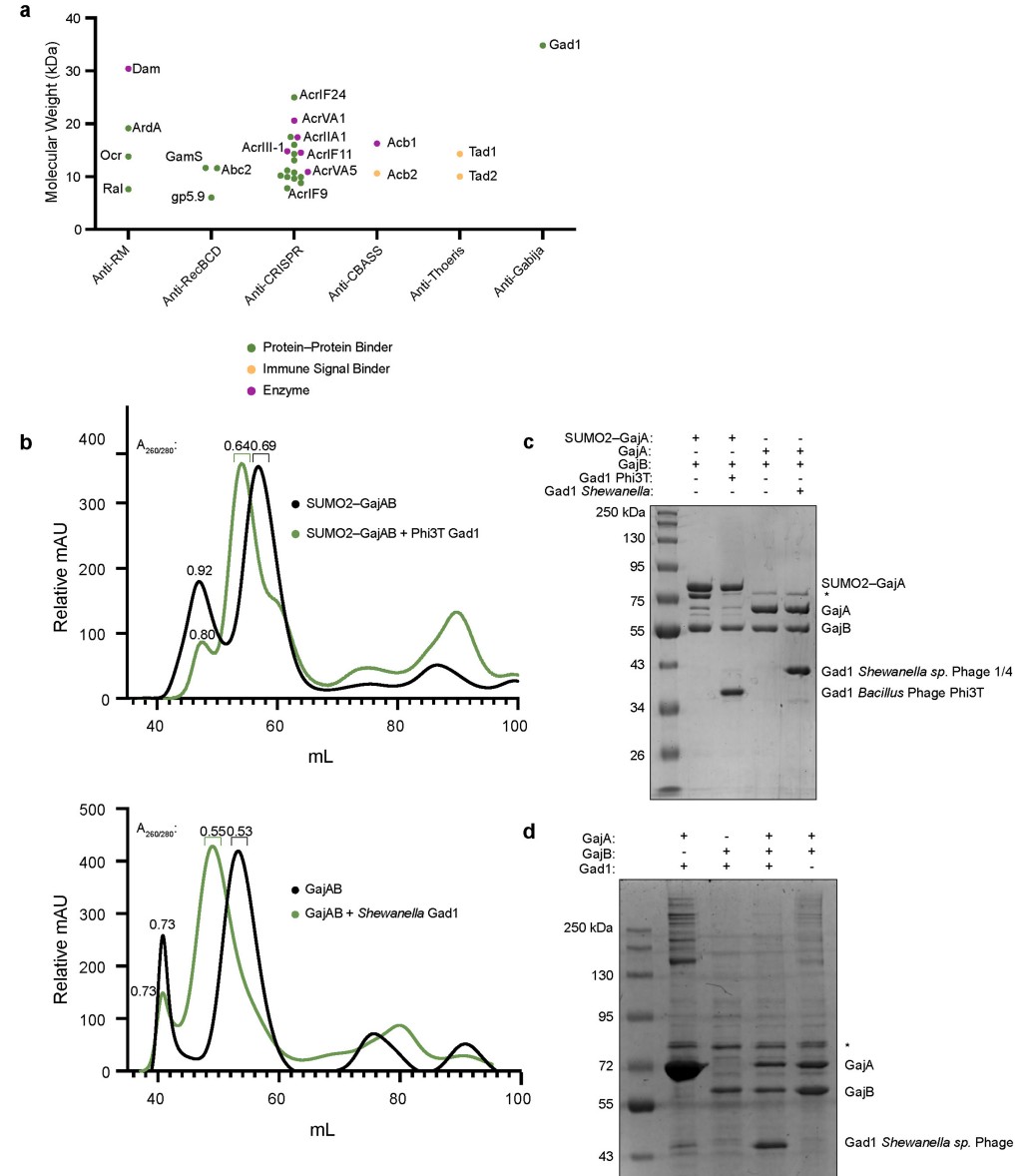

**Extended Data Fig. 4 | Size comparison of Gad1 with known phage immune-evasion proteins and biochemical characterization of Gad1 for binding to the GajAB complex. a**, Analysis of known phage immune-evasion proteins according to function and molecular weight demonstrates that Gad1 is atypically large for an evasion protein that functions through protein–protein interactions with a host anti-phage defence system. Phage immune-evasion proteins are categorized and exhibited as coloured dots coloured according to the key below. Notable evasion proteins are indicated with text labels[24–28,42,50–53]. **b**, Top, size-exclusion chromatography analysis (16/600 S200) of SUMO2-tagged *Bc*GajAB with or without phage Phi3T Gad1 used for cryo-EM structural studies. Bottom, size-exclusion chromatography analysis (16/600 S300) of *Bc*GajAB

with or without *Shewanella* phage 1/4 Gad1 used for biochemical studies. Brackets indicate fractions collected and the $A_{260/280}$ of the final purified proteins is indicated above. *Shewanella* phage 1/4 Gad1 was used preferentially for biochemical studies due to less toxicity during *E. coli* expression. **c**, SDS–PAGE analysis of purified SUMO2-tagged GajAB, SUMO2-tagged GajAB in complex with phage Phi3T Gad1, untagged GajAB, and untagged GajB in complex with *Shewanella* phage 1/4 Gad1. **d**, SDS–PAGE analysis of Ni-NTA co-purified GajA, GajB, and GajAB with *Shewanella* phage 1/4 Gad1 indicates that Gad1 only binds the fully assembled GajAB complex. Asterisk indicates minor contamination with the *E. coli* protein ArnA. Data in **b**–**d** are representative of 3 independent experiments.

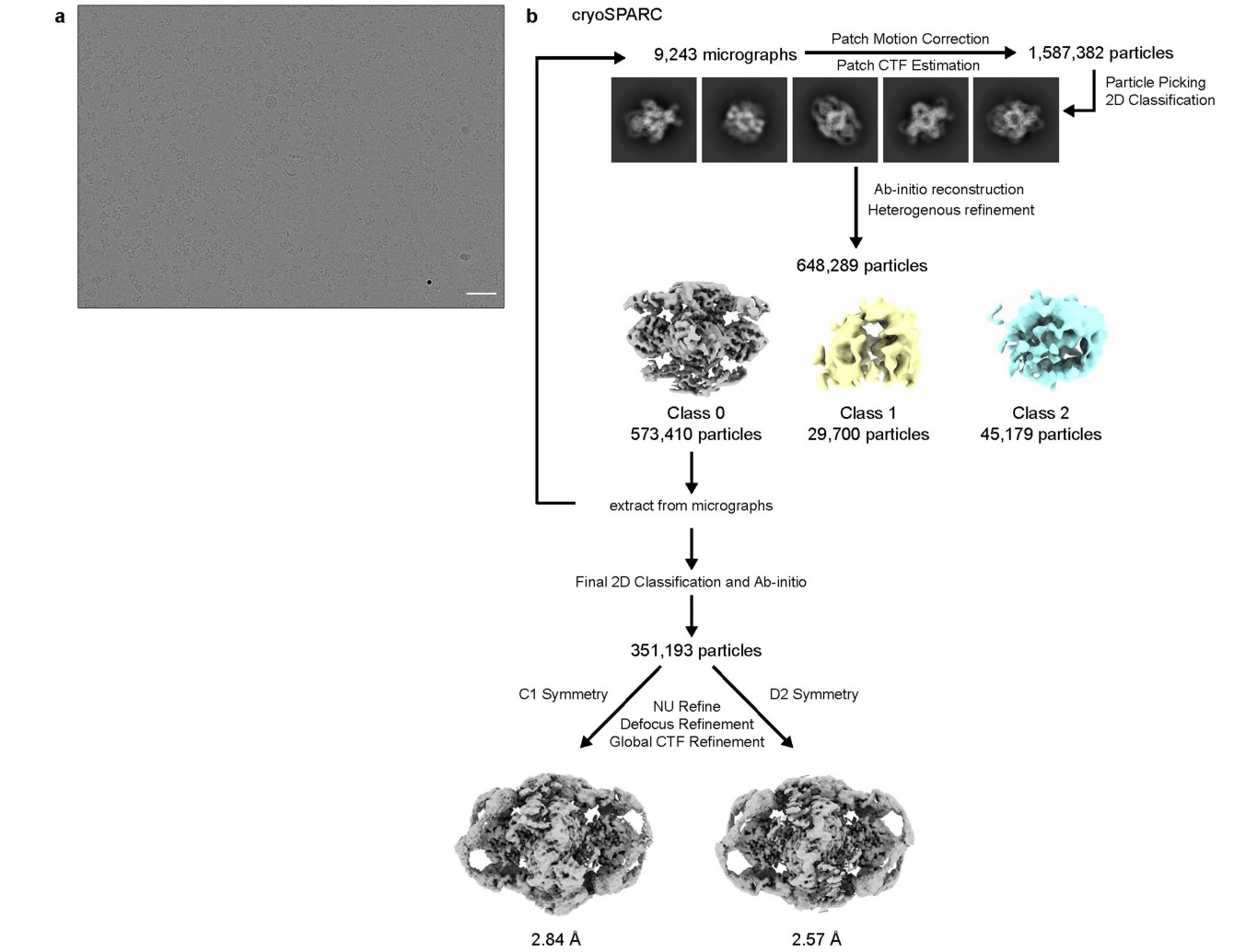

**Extended Data Fig. 5 | Cryo-EM data processing for the GajAB–Gad1 co-complex. a**, Section of a representative electron micrograph (*n* = 9,243) of SUMO2–GajAB in complex with phage Phi3T Gad1. Scale bar, 50 nm. **b**, Data-processing scheme used to generate the final 2.57-Å map.

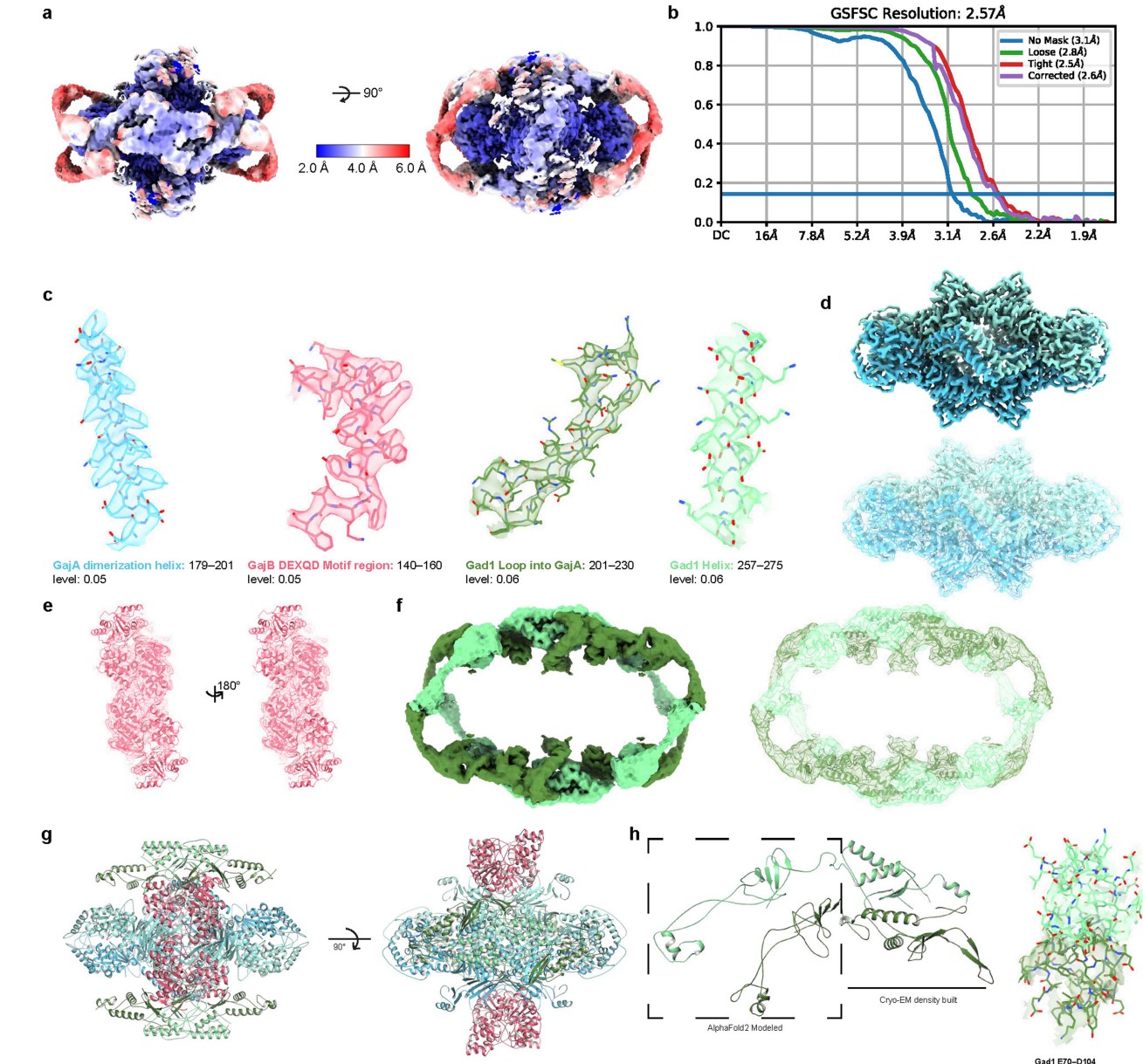

**Extended Data Fig. 6 | Cryo-EM map quality of the GajAB–Gad1 co-complex and model to map fitting. a**, Reconstruction of the GajAB–Gad1 co-complex coloured by local resolution. **b**, Fourier shell correlation (FSC) of the EM map. **c**, GajA, GajB, and Gad1 map to model fit for designated regions. **d**–**f**, Isolated GajA (**d**), GajB (**e**) and Gad1 (**f**) density maps with model fitting. **g**, GajAB–Gad1 model that was used for refining the cryo-EM map for Extended Data Table 2. **h**, Left, sections of Gad1 chains that were built de novo from the cryo-EM density and built using rigid-body placement of AlphaFold2 modelled residues. Right, cryo-EM density used to fit placement of Gad1 fist–fist domain contacts that complete protomer interactions.

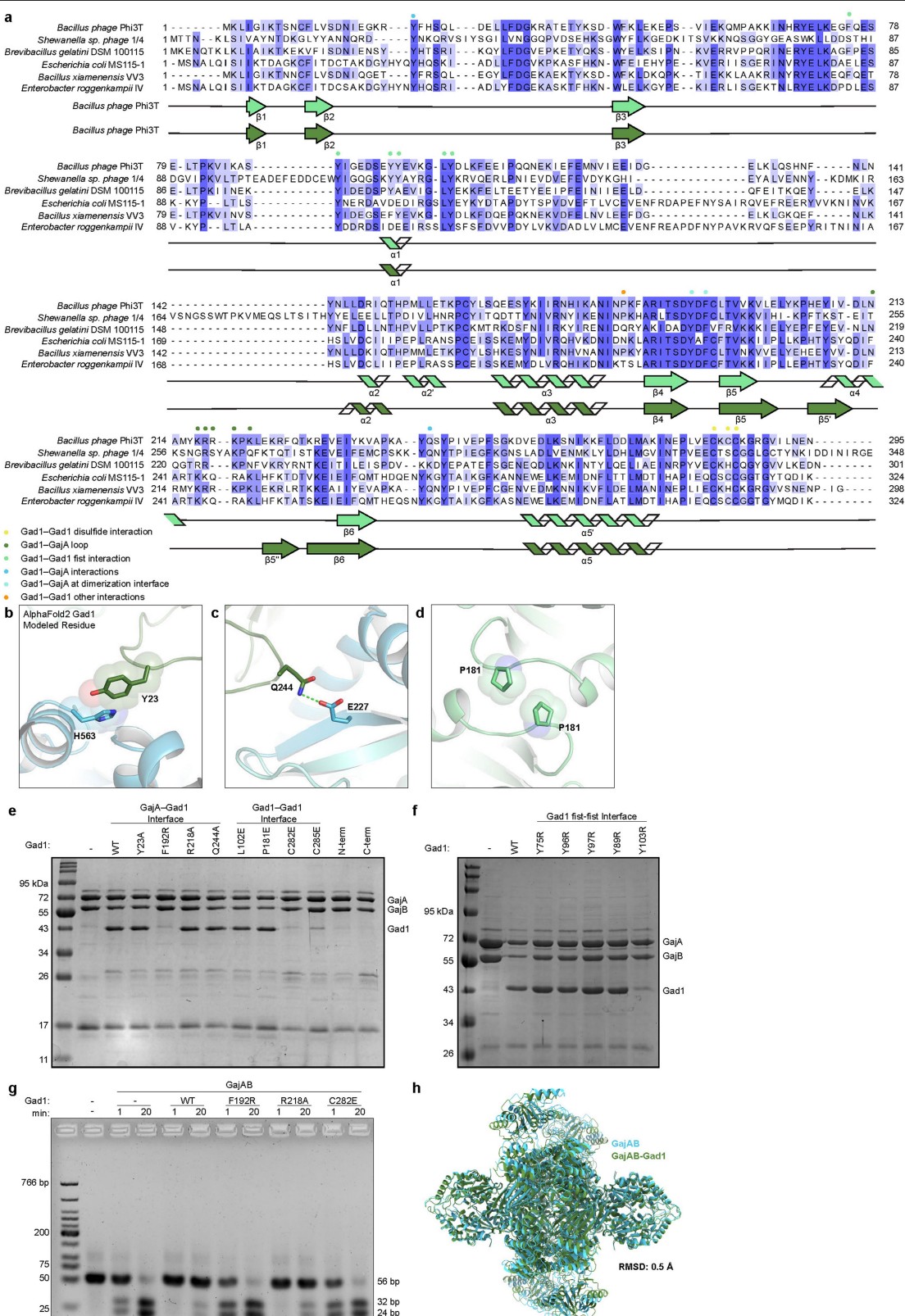

**Extended Data Fig. 7** | See next page for caption.

**Extended Data Fig. 7 | Biochemical and structural characterization of the GajAB–Gad1 co-complex. a**, Structure-guided alignment of Gad1 proteins from indicated phage or prophage genomes coloured according to amino acid conservation. The *Bacillus* phage Phi3T Gad1 secondary structure is displayed according to the two different conformations observed in the GajAB–Gad1 co-complex structure. Oligomerization interface residues are annotated according to the key below. **b,c**, Magnified views of Gad1–GajA interface contacts including hydrophobic interactions in AlphaFold2 arm domain structure of Gad1 and the Toprim domain of GajA (**b**) and Gad1 shoulder domain residue Q244 interaction with GajA dimerization domain residue E277 (**c**). **d**, Magnified view of Gad1–Gad1 oligomerization interactions between shoulder domains of Gad1 protomers. **e**, SDS–PAGE analysis of the ability of *Shewanella* phage 1/4 Gad1 mutant proteins to interact with the GajAB complex. *Shewanella* phage 1/4 Gad1 mutant proteins were co-expressed with SUMO2-tagged GajAB (GajA-tagged) and co-purified by Ni-NTA pull-down. *Shewanella sp.* phage 1/4 Gad1 residues are numbered according to the Phi3T Gad1 structure. To measure high stringency of GajAB–Gad1 interactions, complexes were washed with a 1 M NaCl buffer prior to elution and co-purification. Notably, the Gad1 mutant C282E is no longer able to interact with GajAB in vitro under these stringent conditions, but retains the ability to disrupt Gabija defence in vivo, suggesting that lower-affinity interactions still occur. **f**, SDS–PAGE analysis of the ability of *Shewanella* phage 1/4 Gad1 fist–fist interface mutant proteins to interact with the GajAB complex. *Shewanella* phage 1/4 Gad1 mutant proteins were co-expressed with SUMO2-tagged GajAB (GajA-tagged), co-purified by Ni-NTA pull-down, and treated with SENP2 to cleave the SUMO2 tag prior to SDS–PAGE gel loading. *Shewanella sp.* phage 1/4 Gad1 residues are numbered according to the Phi3T Gad1 structure. **g**, Agarose gel analysis of the ability of GajAB–Gad1 mutant complexes to cleave target 56-bp dsDNA after a 1 min or 20 min incubation. **h**, Superposition of the GajAB crystal structure and GajAB from the GajAB–Gad1 cryo-EM structure demonstrates no significant conformational change after Gad1 binding. Data in **e–g** are representative of 3 independent experiments.

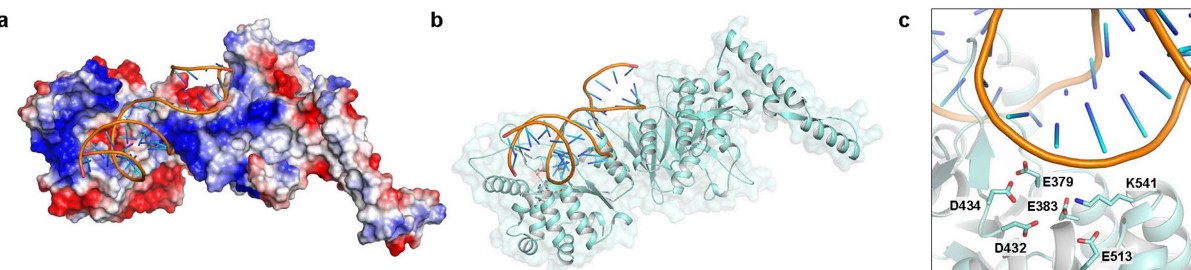

**Extended Data Fig. 8 | Modelling DNA-bound GajA. a,b,** Isolated GajA protomer modelled with DNA bound to the Toprim domain shown with surface electrostatic potential (**a**) and in cartoon format (**b**). DNA modelling was performed using structural homology with the *E. coli* MutS–DNA complex (PDB ID 3K0S)[33]. **c,** Magnified view of the GajA Toprim active site with modelled DNA.

**Extended Data Table 1 | Summary of X-ray data collection, phasing and refinement statistics**

|  | Gabija GajA–GajB (8SM3) |
| --- | --- |
| **Data collection** | |
| Space group | P 6$_2$ 2 2 |
| Cell dimensions | |
| $a$, $b$, $c$ (Å) | 215.79 215.79 173.81 |
| α, β, γ (°) | 90.0, 90.0, 120.0 |
| Resolution (Å) | 49.24–3.00 (3.10–3.00) |
| $R_{pim}$ | 4.0 (80.5) |
| $I / \sigma(I)$ | 15.4 (1.4) |
| Completeness (%) | 100.0 (100.0) |
| Redundancy | 18.1 (16.1) |
|  | |
| **Refinement** | |
| Resolution (Å) | 49.24–3.00 |
| No. reflections | |
| Total | 872109 |
| Unique | 48144 |
| Free | 2000 |
| $R_{work}$ / $R_{free}$ | 23.69 / 26.60 |
| No. atoms | |
| Protein | 8501 |
| Ligand / ion | 5 |
| Water | – |
| B-factors | |
| Protein | 130.19 |
| Ligand / ion | 174.06 |
| Water | – |
| R.m.s. deviations | |
| Bond lengths (Å) | 0.002 |
| Bond angles (°) | 0.442 |

Dataset was collected from an individual crystal. Values in parentheses are for the highest resolution shell.

**Extended Data Table 2 | Cryo-EM data collection, refinement and validation statistics**

|  | GajAB-Gad1 co-complex (EMD-**41983**) (PDB **8U7I**) |
|---|---|
| **Data collection and processing** | |
| Magnification | 37,000 |
| Voltage (kV) | 300 |
| Electron exposure (e–/Å$^2$) | 41.1 |
| Defocus range (μm) | −0.8 to −1.9 |
| Pixel size (Å) | 0.3115 |
| Symmetry imposed | D2 |
| Initial particle images (no.) | 1,587,382 |
| Final particle images (no.) | 351,193 |
| Map resolution (Å) | 2.6 |
| FSC threshold | 0.143 |
| Map resolution range (Å) | 2.56–2.95 |
| | |
| **Refinement** | |
| Initial model used (PDB code) | |
| Model resolution (Å) | 2.57 |
| FSC threshold | 0.143 |
| Model resolution range (Å) | 2.56–2.58 |
| Map sharpening $B$ factor (Å$^2$) | −99.0 |
| Model composition | |
| Non-hydrogen atoms | 38,864 |
| Protein residues | 4,760 |
| Ligands | 0 |
| $B$ factors (Å$^2$) | |
| Proteins | 162.67 |
| R.m.s. deviations | |
| Bond lengths (Å) | 0.004 |
| Bond angles (°) | 0.622 |
| Validation | |
| MolProbity score | 2.06 |
| Clashscore | 10.01 |
| Poor rotamers (%) | 3.07 |
| Ramachandran plot | |
| Favored (%) | 97.06 |
| Allowed (%) | 2.86 |
| Disallowed (%) | 0.09 |

# Reporting Summary

## Statistics

For all statistical analyses, confirm that the following items are present in the figure legend, table legend, main text, or Methods section.

| n/a | Confirmed | |
|---|---|---|
| ☐ | ☒ | The exact sample size (*n*) for each experimental group/condition, given as a discrete number and unit of measurement |
| ☒ | ☐ | A statement on whether measurements were taken from distinct samples or whether the same sample was measured repeatedly |
| ☒ | ☐ | The statistical test(s) used AND whether they are one- or two-sided *Only common tests should be described solely by name; describe more complex techniques in the Methods section.* |
| ☒ | ☐ | A description of all covariates tested |
| ☒ | ☐ | A description of any assumptions or corrections, such as tests of normality and adjustment for multiple comparisons |
| ☐ | ☒ | A full description of the statistical parameters including central tendency (e.g. means) or other basic estimates (e.g. regression coefficient) AND variation (e.g. standard deviation) or associated estimates of uncertainty (e.g. confidence intervals) |
| ☒ | ☐ | For null hypothesis testing, the test statistic (e.g. *F*, *t*, *r*) with confidence intervals, effect sizes, degrees of freedom and *P* value noted *Give P values as exact values whenever suitable.* |
| ☒ | ☐ | For Bayesian analysis, information on the choice of priors and Markov chain Monte Carlo settings |
| ☒ | ☐ | For hierarchical and complex designs, identification of the appropriate level for tests and full reporting of outcomes |
| ☒ | ☐ | Estimates of effect sizes (e.g. Cohen's *d*, Pearson's *r*), indicating how they were calculated |

*Our web collection on statistics for biologists contains articles on many of the points above.*

## Software and code

Policy information about availability of computer code

| Data collection | Serial EM 3.8.6 |
|---|---|
| Data analysis | Coot 0.8.9.2, Phenix 1.20.1, PyMol 2.5.2, cryoSPARC 3.1.018/4.2.0, ChimeraX 1.6.1, AlphaFold v2.2.4, BioRad Quantity One 4.6.9 |

For manuscripts utilizing custom algorithms or software that are central to the research but not yet described in published literature, software must be made available to editors and reviewers. We strongly encourage code deposition in a community repository (e.g. GitHub). See the Nature Portfolio guidelines for submitting code & software for further information.

## Data

Policy information about availability of data

All manuscripts must include a data availability statement. This statement should provide the following information, where applicable:
- Accession codes, unique identifiers, or web links for publicly available datasets
- A description of any restrictions on data availability
- For clinical datasets or third party data, please ensure that the statement adheres to our policy

Coordinates and structure factors of the Gabija GajAB complex have been deposited in PDB under the accession code 8SM3. Coordinates and density maps of the GajAB–Gad1 co-complex are being deposited with PDB and EMDB under accession codes 8U7I and EMDB-41983. All other data are available in the manuscript or the supplementary materials. Source Data are available for Figures 2 and 3.

## Research involving human participants, their data, or biological material

Policy information about studies with human participants or human data. See also policy information about sex, gender (identity/presentation), and sexual orientation and race, ethnicity and racism.

| | |
|---|---|
| Reporting on sex and gender | N/A |
| Reporting on race, ethnicity, or other socially relevant groupings | N/A |
| Population characteristics | N/A |
| Recruitment | N/A |
| Ethics oversight | N/A |

Note that full information on the approval of the study protocol must also be provided in the manuscript.

# Field-specific reporting

Please select the one below that is the best fit for your research. If you are not sure, read the appropriate sections before making your selection.

☒ Life sciences   ☐ Behavioural & social sciences   ☐ Ecological, evolutionary & environmental sciences

For a reference copy of the document with all sections, see nature.com/documents/nr-reporting-summary-flat.pdf

# Life sciences study design

All studies must disclose on these points even when the disclosure is negative.

| | |
|---|---|
| Sample size | No sample size calculations were performed as this is not relevant to biochemical, structural or phage challenge . |
| Data exclusions | No data were excluded from the analyses. |
| Replication | All data were performed with replicates as described. Biochemical or phage challenge experiments were performed with 3 technical replicates and at least 3 independent biological replicates. Figures show one example of a biologically independent replicate and other replicates not shown showed similar results within expected variation. |
| Randomization | Randomization is not relevant to the structural, biochemical, or phage challenge assays described in this study as it would not influence the interpretation of the results. |
| Blinding | Data was not blinded as the data were collected using quantitative measures that were not subjective. |

# Reporting for specific materials, systems and methods

We require information from authors about some types of materials, experimental systems and methods used in many studies. Here, indicate whether each material, system or method listed is relevant to your study. If you are not sure if a list item applies to your research, read the appropriate section before selecting a response.

### Materials & experimental systems

| n/a | Involved in the study |
|---|---|
| ☒ ☐ | Antibodies |
| ☒ ☐ | Eukaryotic cell lines |
| ☒ ☐ | Palaeontology and archaeology |
| ☒ ☐ | Animals and other organisms |
| ☒ ☐ | Clinical data |
| ☒ ☐ | Dual use research of concern |
| ☒ ☐ | Plants |

### Methods

| n/a | Involved in the study |
|---|---|
| ☒ ☐ | ChIP-seq |
| ☒ ☐ | Flow cytometry |
| ☒ ☐ | MRI-based neuroimaging |

