## [Peer Review File · Nature]

Manuscript Title: Structural basis of Gabija anti-phage defence and viral immune evasion

Reviewer Comments & Author Rebuttals

Reviewer Reports on the Initial Version:

Referee expertise:

Referees' comments:

Referee #1 (Remarks to the Author):

Antine et al. determine a structure of the Gabija anti-phage complex GajAB, an inhibitory complex with the phage-encoded Gad1. Overall, this is a strong manuscript with beautiful structures, and this reviewer believes it is novel and would be of interest to general readers. I support publication in Nature, provided GajAB is studied further with additional functional characterization.

Major comments:

1. Since the DNA sequence recognized is known, the authors should determine a structure of GajAB in complex with the target DNA sequence. If they are concerned that the DNA will be cleaved by the complex under the conditions used for cryo-EM (no cleavage kinetics data is included – this should also be added), then the authors should use phosphorothioate-modified DNA substrates. Such a structure would be beneficial to the field, and would increase the impact of this paper.
2. The quality of Gad1 cryo-EM density in the GajAB:Gad1 reconstruction might be improved through further data processing. The map shown in Fig 3b is at a very low density threshold, perhaps it would be better to show the unsharpened map since this is noisy.
3. Since the complex has C2 symmetry, the authors should try symmetry expansion, focused classification and local refinement (potentially with density subtraction if necessary) to improve the quality of these regions. This should improve the quality of the structure, and is important since non-structural biologists may not fully appreciate that regions of Gad1 are not modeled with high confidence.
4. Does Gad1 block DNA binding? This should be tested.

Referee #2 (Remarks to the Author):

Antine et al. report structures of the Gabija anti-phage defense complex, GajAB, and the complex bound by a Gajiba anti-defense protein, Gad1. The GajAB structure is the first reported structure of the complex that mediates Gabija defense, one of the widest spread defense systems in bacteria that was discovered ~5 years ago and remains poorly understood. The structure reveals a large multimeric complex comprising four copies of each protein. The authors demonstrate that GajAB can cleave a previously identified phage DNA sequence in vitro, and that residues identified at interfaces of the GajAB multimer are important for phage defense in vivo. They next solved the cryo-EM structure of the GajAB-Gad1 complex. Gad1 forms an octamer that encircles the GajAB complex. The authors confirm that some interactions between Gad1 and GajAB are important for the inhibitory activity of Gad1, and show that Gad1 inhibits both DNA binding and cleavage in vitro. Finally, the authors model DNA binding to GajA based on a homologous structure, and show that the putative DNA-binding interface is occluded by Gad1 binding.

The structures in this manuscript are impressive and interesting, and provide significant insight into how Gad1 inhibits DNA binding/cleavage by GajA. Although the exact mechanism of GajAB-mediated defense remains unresolved, the structures reported in this manuscript substantially advance our understanding of these systems. I have a few small concerns that the authors could address mostly through changes to the text.

1. Have the authors confirmed that interface mutations tested in Fig. 2f disrupt interactions between subunits? There is a possibility that these mutations may alter phage defense activity through a mechanism other than disruption of complex formation. The authors could provide evidence that some of the mutations disrupt A-B or A-A quaternary structure. If these experiments have not been performed previously or are not possible, the authors could mention the caveat that these mutations have not been confirmed to disrupt the binding interfaces.

2. In ED Fig. 8 it would be helpful to include all of the Gad1 orthologs tested in the accompanying manuscript. The authors of that paper showed that all Gad1 orthologs that could be tested provided anti-Gajiba activity. Are the residues involved in interactions with GajAB conserved in all of these Gad1 variants?

3. On lines 80-85, the authors describe in vitro reconstitution of GajAB cleavage activity. Ref. 22 previously demonstrated DNA cleavage by GajA alone, but not the GajAB complex. The way this section is written suggests that GajAB cleavage of the dsDNA substrate is previously characterized, and that the authors are the first to report cleavage of that substrate by GajA alone. This should be the other way around, as the current manuscript demonstrates that the GajA cleavage that was previously characterized is also observed for the GajAB complex.

4. Line 57-60: Should these sentences reference ED Fig. 1a, which shows the size-exclusion chromatography of GajA and GajB alone, rather than ED Fig. 1b, which shows an SDS-PAGE gel of the

proteins?

5. Line 73: Should Fig. 2e be referenced here, which shows the hydrogen bond between R97 and Q150?

6. In Figure 2f, the labels above the graph are easy to overlook. It might help to move panels e and f further down, away from the panels above, to help emphasize these labels).

7. The authors provide some explanation of why two different Gad1 variants were used for structural and biochemical studies in the figure legend for ED Fig. 5. It would be helpful to expand on this a bit, perhaps in the Methods section. Specifically, were all residues that were mutated in the biochemical experiments conserved between the two variants? This information is available in ED Fig. 8 but could also be mentioned when describing the use of the two constructs.

Referee #3 (Remarks to the Author):

Antine et al present a detailed and solid study on the structure of the Gabija defense system, using a combination of X-ray crystallography and Cryo-EM imaging. In vivo data accompany structural data to inform how Gabija structural determinants impact phage infectivity. Subsequently, the authors performed structural, biochemical and in vivo experiments on a newly identified inhibitor of the Gabija system, Gad1, which is identified and described in the accompanying study by Yirmiya et al. The authors show a highly detailed structure on how Gad1 interacts with the Gabija complex and likely blocks the immune response. The high-resolution structural data are of high quality and very interesting by themselves, but the strength of the paper comes from the complementation of structural data with (1) in vitro binding and cleavage assays and (2) in vivo functionality assays. The biochemical data provide a solid basis for understanding how Gad1 is able to block the Gabija immune response, and the in vivo data help solidify the conclusions (but see my comment below about inconsistencies between in vitro and in vivo data). Overall the paper reads very well and has an excellent logical flow. Experimental designs are solid, data analyses and interpretations are flawless, and figures are mostly highly effective (but see minor comments below). The study details a significant step forward in our understanding of how a commonly found defence provides immunity to its host, and provides exciting new insights in the intricate way a newly discovered counter-defense interferes with immunity. My criticisms to the paper are all minor and can be found below.

Fig. 1b: please indicate which orientation the 3 pictures refer to (e.g. x, y, z plane) to help interpretation of the figure. Same comment for Fig. 3b.

Fig. 1c: It isn't clear what the different subdomains refer to in the GajB monomer structure – please indicate this in Fig. 1a.

Fig. 2. Please indicate (perhaps add it to the inset boxes in Fig. 2?) exactly where the residues are located that were used in the mutational analysis of 2f. Currently only a few of these residues are explicitly mentioned in the inset boxes. Also provide a clearer rationale for how the mutants

selection was done. Same comment for Fig. 3j.

Lines 164-167: There's a discrepancy between in vivo and in vitro data for the C282 Gad1 mutant (strong in vitro effect, which is not reflected in vivo), which is currently not discussed. The authors should address this inconsistency in the discussion of their results.

The complete GabAB-Gad1 complex is truly mindboggling in complexity. The authors speculate very briefly on why phages are using such a complex mechanism for blocking a defence (lines 180-185). Are the authors able to speculate on why the particular features of the GabAB complex (e.g. assembly of the different GabA-GabB components, target binding characteristics) could make it particularly hard for smaller proteins to block the immune response?

Referee #4 (Remarks to the Author):

Gabija is one of the most prevalent anti-phage defense systems in bacteria, and yet the molecular basis of defense remains poorly understood. Here, Antine et al use X-ray crystallography and cryo-EM to define how Gabija proteins assemble into an ~500 kDa supramolecular complex that degrades phage DNA. They go on to show how a phage-encoded protein (Gabija anti-defense 1, Gad1), which is characterized in an accompanying paper, directly assembles into molecular lariat that encircles GajA and inactivates the GajAB complex. The structures are visually striking, mechanistically informative, and collectively reveal a surprising mechanism for a phage encoded suppression systems.

Major suggestions

Please explain if or how Gad1 impacts the structure of GajAB. Consider adding a superposition to the supplement and reporting an RMSD. The superposition should reveal any conformational changes in GajAB, induced by Gad1 binding.

The gel shifts in figure 4C are surprisingly weak. Presumably, this assay is complicated by DNA cleavage activity of the GajAB. Please repeat the gel shifts using EDTA and/or active site mutants that block DNA cleavage, so that DNA binding and Gad1-mediated inhibition can be more clearly demonstrated (required). Consider using labeled substrates and a more quantitative assessment of DNA binding and inhibition activities (not required, but strongly advised).

Minor suggestions

Figure 1b. consider adding cues that help the reader understand how each of the three orientations are related to one another.

Extended Data Figure 1. Please explain what is in the first two major factions of the SEC. Add these to the SDS page and insert the 260/280 ration above each peak. Similar point for Extended Data Figure 5. The large molecular weight peak here is smaller but these are not void peaks and should be discussed.

Extended Data Figure 1. Please provide cleavage motif (as a duplex) used panel c and provide a control where this motif has been scrambled. Without this, readers are forced to find the

appropriate data in the methods and even there the partially palindromic nature of this motif is not easy to identify by eye.

Extended Data Figure 1. It is challenging to see which parts of GajB2B are shared with E. coli Rep 2B, which makes it hard to see how this represents the intermediate state that the authors claim to have identified.

The authors claim “Together, these results define the structural basis of GajA and GajB interaction and demonstrate that GajAB supramolecular complex formation is critical for Gabija anti-phage defense.” I am not sure the phage challenge assays performed using the mutants provide evidence related to “supramolecular complex formation”. Functional importance of residues tested is clear, but data related to assembly or stability is not provided. Consider rephrasing for accuracy.

Author Rebuttals to Initial Comments:

Referee #1:

Antine et al. determine a structure of the Gabija anti-phage complex GajAB, an inhibitory complex with the phage-encoded Gad1. Overall, this is a strong manuscript with beautiful structures, and this reviewer believes it is novel and would be of interest to general readers. I support publication in Nature, provided GajAB is studied further with additional functional characterization.

We are grateful to the reviewer for highlighting our study as a strong manuscript of interest to general readers, and we thank them for their helpful feedback to improve our analysis.

Major comments:

1. Since the DNA sequence recognized is known, the authors should determine a structure of GajAB in complex with the target DNA sequence. If they are concerned that the DNA will be cleaved by the complex under the conditions used for cryo-EM (no cleavage kinetics data is included – this should also be added), then the authors should use phosphorothioate-modified DNA substrates. Such a structure would be beneficial to the field, and would increase the impact of this paper.

Our results demonstrate that GajAB cleaves DNA rapidly with complete degradation occurring within 20 min (see Extended Data Fig. 9c) agreeing with prior kinetic analysis published by Cheng et al (Cheng et al 2021 PMID 33885789). A key challenge is that in the presence of target DNA we observe that GajAB forms a heterogeneous population that migrates as a smear in EMSA experiments. Likely owing to this substantial heterogeneity, we unfortunately have not been able to identify conditions sufficient for stable GajAB–DNA complex assembly and structural analysis. We agree that understanding the structural basis of GajAB DNA recognition is an important goal for the field and we note that our structures of GajAB in the apo and inhibited states provide a new foundation for these future experiments.

2. The quality of Gad1 cryo-EM density in the GajAB:Gad1 reconstruction might be improved through further data processing. The map shown in Fig 3b is at a very low density threshold, perhaps it would be better to show the unsharpened map since this noisy.

We thank the reviewer for their helpful suggestion. We carefully re-analyzed our data and determined that the reconstruction could be further improved upon processing with D2 symmetry. Re-processing yielded a map of improved quality and a 2.57 Å reconstruction after non-uniform refinement with defocus and global CTF refinement. We have completed model-building with the new map and present the updated data including an unsharpened cryo-EM map in Fig. 3b and detailed analysis in Extended Data Fig. 7.

3. Since the complex has C2 symmetry, the authors should try symmetry expansion, focused classification and local refinement (potentially with density subtraction if necessary) to improve the quality of these regions. This should improve the quality of the structure, and is important since non-structural biologists may not fully appreciate that regions of Gad1 are not modeled with high confidence.

We thank the reviewer for these specific suggestions. We performed symmetry expansion and local refinement on the entire map as suggested, however, the quality of the map did not improve. To focus on improving the quality of the Gad1 arm and fist domain map regions, symmetry expansion, particle subtraction, and local refinement were performed on half the complex. Unfortunately, the local refinement did not improve the quality of the Gad1 density (shown below). We believe this is due to inherent flexibility in the Gad1 arm and fist domains that make no significant contacts with the GajA Toprim domain. We have carefully described these features in figures and additionally used docking of AlphaFold2 models to further analyze the Gad1 arm and fist domains (Fig. 3h, Extended Data Fig. 7a,h).

Local Refinement of approximately two GajAs, two GajBs, and four Gad1s to try to improve the resolution of the Gad1 arm and fist domains. Symmetry expansion was used with D2 symmetry to yield 1,404,772 particles. Subtracted volumes were generated using the UCSF Chimera Segger, which were then used in a particle subtraction job to give the desired focus on Gad1 in a local refinement leading to a 6.52 Å reconstruction.

4. Does Gad1 block DNA binding? This should be tested.

We have performed the experiment requested. Our data demonstrate that Gad1 directly inhibits the ability of GajAB to interact with target DNA (see Fig. 4c).

David Taylor and Jack Bravo

We thank the reviewers again for their insights and very helpful suggestions to improve our cryo-EM analysis.

Referee #2:

Antine et al. report structures of the Gabija anti-phage defense complex, GajAB, and the complex bound by a Gajiba anti-defense protein, Gad1. The GajAB structure is the first reported structure of the complex that mediates Gabija defense, one of the widest spread defense systems in bacteria that was discovered ~5 years ago and remains poorly understood. The structure reveals a large multimeric complex comprising four copies of each protein. The authors demonstrate that GajAB can cleave a previously identified phage DNA sequence in vitro, and that residues identified at interfaces of the GajAB multimer are important for phage defense in vivo. They next solved the cryo-EM structure of the GajAB-Gad1 complex. Gad1 forms an octamer that encircles the GajAB complex. The authors confirm that some interactions between Gad1 and GajAB are important for the inhibitory activity of Gad1, and show that Gad1 inhibits both DNA binding and cleavage in vitro. Finally, the authors model DNA binding to GajA based on a homologous structure, and show that the putative DNA-binding interface is occluded by Gad1 binding.

The structures in this manuscript are impressive and interesting, and provide significant insight into how Gad1 inhibits DNA binding/cleavage by GajA. Although the exact mechanism of GajAB-mediated defense remains unresolved, the structures reported in this manuscript substantially advance our understanding of these systems. I have a few small concerns that the authors could address mostly through changes to the text.

We appreciate the reviewer for highlighting our work as impressive and providing significant insight into understanding of Gabija anti-phage defense. We thank them for their very helpful comments.

1. Have the authors confirmed that interface mutations tested in Fig. 2f disrupt interactions between subunits? There is a possibility that these mutations may alter phage defense activity through a mechanism other than disruption of complex formation. The authors could provide evidence that some of the mutations disrupt A-B or A-A quaternary structure. If these experiments have not been performed previously or are not possible, the authors could mention the caveat that these mutations have not been confirmed to disrupt the binding interfaces.

To address the reviewer's point, our revised manuscript now includes new pull-down experiments to monitor the impact of mutations in the GajA–GajB interfaces on Gabija complex assembly. GajA (K94E, R97A) and GajB (V147E, Q150R) mutations to the GajA–GajB interface disrupt interactions *in vitro*, confirming that these interfaces are specifically required for complex assembly (Extended Data Fig. 1f). We additionally tested mutations to the GajA–GajA (L191E, I199E, K229E, H295R, R139E, D135R) and GajB–GajB (I122E, N121R) interfaces and observed that these mutations do not impact GajA–GajB interaction in this assay, however, we have added a line in the Extended Data Fig. 1 legend to state that it is not known if these mutants remain competent at forming the wildtype 4:4 complex. In some cases, these mutants resulted in lower expression levels suggesting they may also disrupt overall protein stability. Together these results provide further evidence that GajA–GajB assembly is critical for productive Gabija anti-phage defense.

2. In ED Fig. 8 it would be helpful to include all of the Gad1 orthologs tested in the accompanying manuscript. The authors of that paper showed that all Gad1 orthologs that could be tested provided anti-Gajiba activity. Are the residues involved in interactions with GajAB conserved in all of these Gad1 variants?

We thank the reviewer and have updated the Gad1 alignment as requested to include all homologs from the Yirmiya et al study (see revised Extended Data Fig. 8a). The updated alignment demonstrates high conservation of most Gad1 residues that form Gad1–GajA and Gad1–Gad1 contacts in our structure. Notably, the alignment reveals high conservation of the region around F192, which is critical for Gad1 to contact the GajA dimerization domain (Fig. 3f,j; Extended Data Fig. 8a), further supporting that this region is critical for inhibition of Gabija anti-phage defense.

3. On lines 80-85, the authors describe *in vitro* reconstitution of GajAB cleavage activity. Ref. 22 previously demonstrated DNA cleavage by GajA alone, but not the GajAB complex. The way this section is written suggests that GajAB cleavage of the dsDNA substrate is previously characterized, and that the authors are the first to report cleavage of that substrate by GajA alone. This should be the other way around, as the current manuscript demonstrates that the GajA cleavage that was previously characterized is also observed for the GajAB complex.

We agree with the reviewer and have corrected this text to make sure the important prior work by Cheng et al is clear: “We reconstituted Gabija activity *in vitro* and observed that the GajAB complex binds and rapidly cleaves a previously characterized 56 bp dsDNA substrate containing a sequence specific motif derived from phage lambda DNA (Extended Data Fig. 1c) (Cheng et al 2021 PMID 33885789). The GajAB complex can interact with a scrambled DNA sequence but is unable to cleave this target DNA (Extended Data Fig. 1c,d). GajA and GajB proteins are each essential for phage defense *in vivo* (Doron et al 2018 PMID 29371424; Cheng et al 2021 PMID 33885789), but we observed in agreement with previous results that GajA is alone sufficient to cleave target DNA and does not require GajB *in vitro* (Extended Data Fig. 1c) (Cheng et al 2021 PMID 33885789; Cheng et al 2023 PMID 37480847). These results suggest GajAB complex formation may have a specific role in controlling substrate recognition or nuclease activation during phage infection.” (see Lines 79–87).

4. Line 57-60: Should these sentences reference ED Fig. 1a, which shows the size-exclusion chromatography of GajA and GajB alone, rather than ED Fig. 1b, which shows an SDS-PAGE gel of the proteins?

We thank the reviewer for noting the mistakes in points 4 and 5. We have corrected the text to reference Extended Data Fig. 1a.

5. Line 73: Should Fig. 2e be referenced here, which shows the hydrogen bond between R97 and Q150?

We have corrected the text to reference Fig 2e.

6. In Figure 2f, the labels above the graph are easy to overlook. It might help to move panels e and f further down, away from the panels above, to help emphasize these labels).

We thank the reviewer for this suggestion. We have moved panels e and f down to allow for more emphasis of the labels (see revised Fig. 2).

7. The authors provide some explanation of why two different Gad1 variants were used for structural and biochemical studies in the figure legend for ED Fig. 5. It would be helpful to expand on this a bit, perhaps in the Methods section. Specifically, were all residues that were mutated in the biochemical experiments conserved between the two variants? This information is available in ED Fig. 8 but could also be mentioned when describing the use of the two constructs.

We thank the reviewer for this suggestion. We have added the following sentences to the “Protein expression and purification” methods section. “Co-expression of Gabija with Phi3T Gad1 results in mild toxicity in *E. coli* grown on MDG media plates. No toxicity was observed using a closely related Gad1 homolog from the *Shewanella* phage 1/4. Biochemical analysis of Gabija–Gad1 interactions was therefore conducted with *Shewanella* phage 1/4 Gad1. Notably, all Gad1 residues analyzed are 100% conserved between Phi3T Gad1 and *Shewanella* phage 1/4 Gad1.” (see Lines 420–424).

Referee #3:

Antine et al present a detailed and solid study on the structure of the Gabija defense system, using a combination of X-ray crystallography and Cryo-EM imaging. In vivo data accompany structural data to inform how Gabija structural determinants impact phage infectivity. Subsequently, the authors performed structural, biochemical and in vivo experiments on a newly identified inhibitor of the Gabija system, Gad1, which is identified and described in the accompanying study by Yirmiya et al. The authors show a highly detailed structure on how Gad1 interacts with the Gabija complex and likely blocks the immune response. The high-resolution structural data are of high quality and very interesting by themselves, but the strength of the paper comes from the complementation of structural data with (1) in vitro binding and cleavage assays and (2) in vivo functionality assays. The biochemical data provide a solid basis for understanding how Gad1 is able to block the Gabija immune response, and the in vivo data help solidify the conclusions (but see my comment below about inconsistencies between in vitro and in vivo data). Overall the paper reads very well and has an excellent logical flow. Experimental designs are solid, data analyses and interpretations are flawless, and figures are mostly highly effective (but see minor comments below). The study details a significant step forward in our understanding of how a commonly found defence provides immunity to its host, and provides exciting new insights in the intricate way a newly discovered counter-defense interferes with immunity. My criticisms to the paper are all minor and can be found below.

We thank the reviewer for their enthusiasm for our study and are grateful for their helpful feedback to improve our manuscript.

Fig. 1b: please indicate which orientation the 3 pictures refer to (e.g. x, y, z plane) to help interpretation of the figure. Same comment for Fig. 3b.

We thank the reviewer for their suggestion. We have now added the rotation arrows necessary to generate the structure orientations in Fig. 1b and Fig. 3b.

Fig. 1c: It isn't clear what the different subdomains refer to in the GajB monomer structure – please indicate this in Fig. 1a.

To improve clarity of this figure, we have added the GajB subdomains to the linear GajB diagram in Fig. 1a.

Fig. 2. Please indicate (perhaps add it to the inset boxes in Fig. 2?) exactly where the residues are located that were used in the mutational analysis of 2f. Currently only a few of these residues are explicitly mentioned in the inset boxes. Also provide a clearer rationale for how the mutants selection was done. Same comment for Fig. 3j.

We thank the reviewer for this suggestion. Due to figure size restraints, all the point mutations that were used for phage defense assays in Fig. 2f and Fig. 3j do not fit within the representative figures. We have now added the missing interactions to Extended Data Fig. 2b,c for GajA protomer interactions and Extended Data Fig. 8b–d for Gad1–GajA and Gad1–Gad1 interactions. We also added the following sentence to the legend of Figure 2 “GajA and GajB mutations were selected by identifying central residues with well-defined protein–protein contacts within each multimerization interface and tested to determine the impact on the ability of the *B. cereus* Gabija operon to defend cells against phage infection.” (see Lines 310–313).

Lines 164-167: There's a discrepancy between in vivo and in vitro data for the C282 Gad1 mutant (strong in vitro effect, which is not reflected in vivo), which is currently not discussed. The authors should address this inconsistency in the discussion of their results.

We thank the reviewer for this point. We have now added to the legend of Extended Data Fig. 9 the sentences “To measure high stringency of GajAB–Gad1 interactions, complexes were washed with a 1 M NaCl buffer prior to elution and co-purification. Notably, the Gad1 mutant C282E is no longer able to interact with GajAB in vitro under these stringent conditions, but retains the ability to disrupt Gabija defense *in vivo* suggesting that lower affinity interactions still occur.” (see Lines 667–671).

The complete GabAB-Gad1 complex is truly mindboggling in complexity. The authors speculate very briefly on why phages are using such a complex mechanism for blocking a defence (lines 180-185). Are the authors able to speculate on why the particular features of the GabAB complex (e.g. assembly of the different GabA-GabB components, target binding characteristics) could make it particularly hard for smaller proteins to block the immune response?

We agree with the reviewer that the complexity of Gad1 is particularly surprising. Due to the diversity of mechanisms phages are known to use to subvert host immunity we are hesitant to speculate more on why smaller anti-Gabija proteins may not be sufficient, but we hope that our study encourages other groups to continue to identify additional inhibitors to help further understand Gabija function.

Referee #4:

Gabija is one of the most prevalent anti-phage defense systems in bacteria, and yet the molecular basis of defense remains poorly understood. Here, Antine et al use X-ray crystallography and cryo-EM to define how Gabija proteins assemble into an ~500 kDa supramolecular complex that degrades phage DNA. They go on to show how a phage-encoded protein (Gabija anti-defense 1, Gad1), which is characterized in an accompanying paper, directly assembles into molecular lariat that encircles GajA and inactivates the GajAB complex. The structures are visually striking, mechanistically informative, and collectively reveal a surprising mechanism for a phage encoded suppression systems.

We appreciate the reviewer for their positive comments for our new structures and thank them for their helpful critique to improve our study.

Major suggestions

Please explain if or how Gad1 impacts the structure of GajAB. Consider adding a superposition to the supplement and reporting an RMSD. The superposition should reveal any conformational changes in GajAB, induced by Gad1 binding.

We thank the reviewer for this suggestion. We superposed our GajAB crystal structure with GajAB from the GajAB–Gad1 cryo-EM structure and observed no significant changes (α -carbon RMSD of 0.5 Å). We now provide these data as a new panel in Extended Data Fig. 9d and include this point in the main text (see Lines 162–164).

The gel shifts in figure 4C are surprisingly weak. Presumably, this assay is complicated by DNA cleavage activity of the GajAB. Please repeat the gel shifts using EDTA and/or active site mutants that block DNA cleavage, so that DNA binding and Gad1-mediated inhibition can be more clearly demonstrated (required). Consider using labeled substrates and a more quantitative assessment of DNA binding and inhibition activities (not required, but strongly advised).

A key challenge is that in the presence of target DNA we observe that GajAB forms a heterogeneous population that migrates as a smear in our EMSA experiments. We interpret these data to suggest not that DNA binding is weak, but that DNA binding results in heterogenous complex formation. To address this point further, we performed EMSA experiments with catalytically inactive Toprim domain GajAB complexes (GajA [E379A]–GajB) as suggested and observed that complex formation still results in creation of a heterogenous population. As additional biochemical experiments in our revised manuscript, we tested a scrambled DNA sequence as requested (see below) and observed that the GajAB complex is unable to cleave this DNA but retains the ability to form a GajAB–DNA complex (Extended Data Fig. 1d). Our study provides a foundation for future experiments to further define the structural basis of Gabija–DNA complex assembly.

Minor suggestions

Figure 1b. consider adding cues that help the reader understand how each of the three orientations are related to one another.

We thank the reviewer for their suggestion. We have now added the rotation arrows necessary to generate the structure orientations in Fig. 1b and Fig. 3b.

Extended Data Figure 1. Please explain what is in the first two major factions of the SEC. Add these to the SDS page and insert the 260/280 ration above each peak. Similar point for Extended Data Figure 5. The large molecular weight peak here is smaller but these are not void peaks and should be discussed.

All of our data support a model where GajA forms a core homo-tetrameric complex and that docking of GajB results in a 4:4 assembly. In our S200 size-exclusion purification of Gabija proteins, we observed that GajA migrates as stable tetramer species and a larger heterogenous population that is closer to the void of the column (Extended Data Fig. 1a). GajB migrates as stable monomeric species, and in the presence of GajA we observe that both proteins shift and form a dominant species that our structural data demonstrate is the complete 4:4 GajAB assembly. In our S300 size-exclusion purification of the GajAB–Gad1 complex we see that co-expression with Gad1 results in formation of a larger stable assembly that our structural analysis demonstrates is an 8:4:4 complex. None of these populations have significant nucleic acid as judged by the A260/280 ratios and for clarity we have added these ratios above the graphs in Extended Data Fig. 1c and 5a.

We do not have a biochemical or structural explanation for the identity of the larger GajA species, but this population is unstable, and we suspect may be a result of GajA being expressed in the absence of the partnering GajB protein necessary to complete assembly. We observed that the GajA tetramer species is sufficient for target DNA cleavage *in vitro* (Extended Data Fig. 1c) but we specifically focused our analysis on the GajAB complex as both proteins are essential for anti-phage defense *in vivo*.

Extended Data Figure 1. Please provide cleavage motif (as a duplex) used panel c and provide a control where this motif has been scrambled. Without this, readers are forced to find the appropriate data in the methods and even there the partially palindromic nature of this motif is not easy to identify by eye.

We thank the reviewer for this helpful suggestion. To address this point, we now include additional biochemical data testing Gabija DNA cleavage activity against a scrambled version of the target DNA sequence. These results demonstrate that the GajAB complex is able to bind scrambled DNA but is unable to cleave this DNA as a target substrate (Extended Data Fig. 1c,d). We additionally now include the target DNA sequence derived from Cheng et al (Cheng et al 2021 PMID 33885789) and the scrambled DNA sequence within Extended Data Fig. 1c.

Extended Data Figure 1. It is challenging to see which parts of GajB2B are shared with E. coli Rep 2B, which makes it hard to see how this represents the intermediate state that the authors claim to have identified.

We apologize for the challenging nature of our presentation. We have remade the structure diagrams of GajB and *E. coli* Rep to highlight only the 2B domain (Extended Data Fig. 1e). The 2B domain are

colored in blue for Rep and brown for GajB highlighting how the GajB 2B domain is in an intermediate conformation relative to the inactive and active Rep 2B domains.

The authors claim “Together, these results define the structural basis of GajA and GajB interaction and demonstrate that GajAB supramolecular complex formation is critical for Gabija anti-phage defense.” I am not sure the phage challenge assays performed using the mutants provide evidence related to “supramolecular complex formation”. Functional importance of residues tested is clear, but data related to assembly or stability is not provided. Consider rephrasing for accuracy.

To address the reviewer’s point, our revised manuscript now includes new pull-down experiments to monitor the impact of mutations in the GajA–GajB interfaces on Gabija complex assembly. GajA (K94E, R97A) and GajB (V147E, Q150R) mutations to the GajA–GajB interface disrupt interactions *in vitro*, confirming that these interfaces are specifically required for complex assembly (Extended Data Fig. 1f). We additionally tested mutations to the GajA–GajA (L191E, I199E, K229E, H295R, R139E, D135R) and GajB–GajB (I122E, N121R) interfaces and observed that these mutations do not impact GajA–GajB interaction in this assay, however, we have added a line in the Extended Data Fig. 1 legend to state that it is not known if these mutants remain competent at forming the wildtype 4:4 complex. In some cases, these mutants resulted in lower expression levels suggesting they may also disrupt overall protein stability. Together these results provide further evidence that GajA–GajB assembly is critical for productive Gabija anti-phage defense.

Reviewer Reports on the First Revision:

Referees' comments:

Referee #1 (Remarks to the Author):

The authors have done a more than satisfactory job addressing our initial suggestions/comments. Furthermore, they have significantly improved the GajAB:Gad1 reconstruction and modeling, leading to further insights. This reviewer strongly supports publication in Nature. The mechanism of the Gabija anti-phage complex is novel and exciting and would be of interest to a broad audience.

David Taylor and Jack Bravo

Referee #2 (Remarks to the Author):

The authors have addressed my concerns, and I believe they have done an excellent job of responding to the other reviewers' concerns as well. I believe this manuscript is suitable for publication as is.

Referee #3 (Remarks to the Author):

The authors have done substantial work to address the reviewers' comments. the majority of the requested extra experiments and /or analyses has been done to a satisfactory level. The request for solving an additional structure of GajAB with target DNA bound to it (Reviewer 1) has not been performed, but in my view the authors have rebutted this in a satisfactory manner. Overall, I am happy with the changes made by the authors.

Referee #4 (Remarks to the Author):

My concerns have been addressed and I support the publication of this work in Nature.

Author Rebuttals to First Revision:

Referee #1:

The authors have done a more than satisfactory job addressing our initial suggestions/comments. Furthermore, they have significantly improved the GajAB:Gad1 reconstruction and modeling, leading to further insights. This reviewer strongly supports publication in Nature. The mechanism of the Gabija anti-phage complex is novel and exciting and would be of interest to a broad audience.

David Taylor and Jack Bravo

We thank the reviewers again for their very helpful feedback.

Referee #2:

The authors have addressed my concerns, and I believe they have done an excellent job of responding to the other reviewers' concerns as well. I believe this manuscript is suitable for publication as is.

We thank the reviewer again for their very helpful feedback.

Referee #3:

The authors have done substantial work to address the reviewers' comments. the majority of the requested extra experiments and /or analyses has been done to a satisfactory level. The request for solving an additional structure of GajAB with target DNA bound to it (Reviewer 1) has not been performed, but in my view the authors have rebutted this in a satisfactory manner. Overall, I am happy with the changes made by the authors.

We thank the reviewer for their helpful feedback and for agreeing that the structure of GajAB in complex with DNA is beyond the scope of this manuscript.

Referee #4

My concerns have been addressed and I support the publication of this work in Nature.

We thank the reviewer again for their very helpful feedback.